# A randomized feasibility trial comparing four antimalarial drug regimens to induce *Plasmodium falciparum* gametocytemia in the controlled human malaria infection model

Isaie J Reuling[1], Lisanne A van de Schans[1], Luc E Coffeng[2], Kjerstin Lanke[1], Lisette Meerstein-Kessel[1], Wouter Graumans[1], Geert-Jan van Gemert[1], Karina Teelen[1], Rianne Siebelink-Stoter[1], Marga van de Vegte-Bolmer[1], Quirijn de Mast[3], André J van der Ven[3], Karen Ivinson[4], Cornelus C Hermsen[1], Sake de Vlas[2], John Bradley[5], Katharine A Collins[6], Christian F Ockenhouse[4], James McCarthy[6], Robert W Sauerwein[1†], Teun Bousema[1†]*

[1]Department of Medical Microbiology, Radboud university medical center, Nijmegen, Netherlands; [2]Department of Public Health, Erasmus MC, University Medical Center Rotterdam, Rotterdam, Netherlands; [3]Department of Internal Medicine, Radboud University Medical Center, Nijmegen, Netherlands; [4]PATH Malaria Vaccine Initiative, Washington, United States; [5]MRC Tropical Epidemiology Group, London School of Hygiene and Tropical Medicine, London, United Kingdom; [6]Clinical Tropical Medicine Laboratory, QIMR Berghofer, Brisbane, Australia

*For correspondence:
teun.bousema@radboudumc.nl

†These authors contributed equally to this work

Competing interests: The authors declare that no competing interests exist.

## Abstract

**Background:** Malaria elimination strategies require a thorough understanding of parasite transmission from human to mosquito. A clinical model to induce gametocytes to understand their dynamics and evaluate transmission-blocking interventions (TBI) is currently unavailable. Here, we explore the use of the well-established Controlled Human Malaria Infection model (CHMI) to induce gametocyte carriage with different antimalarial drug regimens.

**Methods:** In a single centre, open-label randomised trial, healthy malaria-naive participants (aged 18–35 years) were infected with Plasmodium falciparum by bites of infected Anopheles mosquitoes. Participants were randomly allocated to four different treatment arms (n = 4 per arm) comprising low-dose (LD) piperaquine (PIP) or sulfadoxine-pyrimethamine (SP), followed by a curative regimen upon recrudescence. Male and female gametocyte densities were determined by molecular assays.

**Results:** Mature gametocytes were observed in all participants (16/16, 100%). Gametocytes appeared 8.5–12 days after the first detection of asexual parasites. Peak gametocyte densities and gametocyte burden was highest in the LD-PIP/SP arm, and associated with the preceding asexual parasite biomass (p=0.026). Male gametocytes had a mean estimated circulation time of 2.7 days (95% CI 1.5–3.9) compared to 5.1 days (95% CI 4.1–6.1) for female gametocytes. Exploratory mosquito feeding assays showed successful sporadic mosquito infections. There were no serious adverse events or significant differences in the occurrence and severity of adverse events between study arms (p=0.49 and p=0.28).

**Conclusions:** The early appearance of gametocytes indicates gametocyte commitment during the first wave of asexual parasites emerging from the liver. Treatment by LD-PIP followed by a curative SP regimen, results in the highest gametocyte densities and the largest number of gametocyte-positive days. This model can be used to evaluate the effect of drugs and vaccines on gametocyte

dynamics, and lays the foundation for fulfilling the critical unmet need to evaluate transmission-blocking interventions against falciparum malaria for downstream selection and clinical development.

**Funding:** Funded by PATH Malaria Vaccine Initiative (MVI).

**Clinical trial number:** NCT02836002.

## Introduction

Malaria, a disease caused by *Plasmodium* parasites, continues to be a public health burden. Despite a reduction in the malaria case incidence of ~40%, and mortality by 62% over the last decade, malaria caused ~429,000 deaths in 2015 (*World Health Organization, 2016*). Apart from the direct health implications, malaria is a substantial contributor to ongoing poverty in affected countries. Recently, the spread of artemisinin-resistant parasites has emerged as a global health concern. Both the recent gains in malaria control and concerns about artemisinin resistance have stimulated programs to eliminate malaria (*World Health Organization, 2016*). Novel interventions may support malaria elimination efforts in endemic settings (*Griffin et al., 2010*) that are further dependent on political and financial commitments to maximize coverage with currently available interventions and improve surveillance systems to optimize disease notification and treatment (*Moonen et al., 2010*).

A major challenge to eliminating malaria is its highly efficient transmission by *Anopheles* mosquitoes. Transmission to mosquitoes starts when a small proportion of asexual parasites commit to form male and female gametocytes. It is currently unclear what stimulates gametocyte commitment and when gametocyte commitment first occurs (*Nilsson et al., 2015*). Upon commitment, maturation of gametocytes takes place predominantly in the bone marrow, and requires 7 days (range 4–12) of development. (*Eichner et al., 2001*) Subsequently, mature gametocytes (parasites that are not associated with clinical disease) appear in the peripheral blood, where they may circulate for an average of 6 days (*Eichner et al., 2001*; *Bousema et al., 2010*). During this period, blood-feeding *Anophelines* may ingest gametocytes where, after a sporogonic development phase, sporozoites reach the mosquito salivary gland rendering the mosquito infectious to humans upon its next bite. Early work based on the microscopic evaluation of experimental *P. falciparum* infection (malariatherapy) studies reported that gametocytes may make their appearance in small numbers around 10 days following the first day of fever (*Shute and Maryon, 1951*; *Ciuca et al., 1937*).

The renewed focus on malaria elimination requires a thorough understanding of malaria transmission dynamics - when mature male and female gametocytes are first produced upon infection and how long they circulate in peripheral blood (*Sinden, 2017*). These parameters are difficult to measure in naturally acquired infections where frequent super-infections, immunity and other factors dictate parasite and gametocyte dynamics (*Bousema and Drakeley, 2011*). Interventions that specifically aim to reduce gametocyte development, circulation time or infectivity are highly desirable in the context of malaria elimination and require effective models for the early clinical evaluation.

The controlled human malaria infection (CHMI) model allows the induction of parasitemia under highly standardized conditions and plays an important role in the assessment of safety and efficacy of novel antimalarial drugs and vaccines (*Sauerwein et al., 2011*). Preliminary evidence for the induction of female gametocytes in CHMI studies with blood stage inoculum was recently demonstrated using piperaquine monotherapy (*Pasay et al., 2016*; *Farid et al., 2017*).

In this study, we aimed to develop a CHMI transmission model to induce gametocyte carriage after mosquito bite infection. The primary objective of the current trial was to safely induce gametocytemia in study participants by the use of different (sub)curative drug regimens based on sulfadoxine-pyrimethamine (*Bousema and Drakeley, 2011*; *Butcher, 1997*) and piperaquine (*Adjalley et al., 2011*).

## Results

From a total of 49 screened candidate participants, 16 volunteers were included in a first cohort and randomly assigned to four study arms prior to challenge (*Figure 1*). After observed transient liver enzyme elevations in the first cohort, the study was temporarily put on hold and the already initiated

**eLife digest** The parasite that causes malaria, named *Plasmodium falciparum*, has a life cycle that involves both humans and mosquitoes. Starting in the saliva of female Anopheles mosquitoes, it enters a person's bloodstream when the insects feed. It then moves to the person's liver, where it infects liver cells and matures into a stage known as schizonts. The schizonts then divide to form thousands of so-called merozoites, which burst out of the liver cells and into the bloodstream. The merozoites infect red blood cells, producing more schizonts and yet more merozoites, which continue the infection.

To complete its life cycle, the parasite must return to a mosquito. Some of the parasites in the person's blood transform into male and female cells called gametocytes that are taken up by a mosquito when it feeds on that person. Inside the mosquito, male and female parasites reproduce to create the next generation of parasites. The new parasites then move to the mosquito's salivary glands, ready to begin another infection. Stopping the parasite being transmitted from humans to mosquitoes will stop the spread of malaria in the population. Yet it has proven difficult to study this part of the life cycle from natural infections.

Here, Reuling et al. report a new method for generating gametocytes in human volunteers that will enable closer study of the biology of malaria transmission. The method is developed using the Controlled Human Malaria Infection (CHMI) model. Healthy volunteers without a history of malaria are bitten by mosquitoes infected with malaria parasites. Shortly afterwards, the volunteers are given a drug treatment to control and reduce their symptoms. The gametocytes form during this phase of the infection. At the end of the experiment, all the volunteers receive a final treatment that completely cures the infection.

Reuling et al. recruited 16 volunteers and assigned them to four groups at random. Each group received a different drug regime. Roughly a week after the mosquito bites, all participants showed malaria parasites in their blood, and between 8.5 and 12 days later, mature gametocytes started to appear. This early appearance suggests that the parasites start to transform into gametocytes when they first emerge from the liver. The experiment also revealed that female gametocytes stay in the blood for a longer period than their male counterparts.

These results are proof of principle for a new way to investigate malaria infection. The new model provides a controlled method for studying *P. falciparum* gametocytes in people. In the future, it could help to test the impact of drugs and vaccines on gametocytes. Understanding more about these parasites' biology could lead to treatments that block malaria transmission.

infections in the second cohort of 13 participants were abrogated by curative treatment on day 3 post challenge. The hold was lifted after reviewing safety data. Participants from the first cohort completed all study visits, and form the basis of the current manuscript; their baseline characteristics are shown in *Table 1*. After exposure to bites of a standard protocol of five *P. falciparum* infected mosquitoes, all participants developed parasitemia on days 6.5–12 post-challenge; peak parasite densities ranged from 1050 to 63113 *Pf*/mL (*Figure 2*; *Figure 2—figure supplement 1*; *Table 2*; *Supplementary file 1*). Due to asexual recrudescence in seven of the eight participants after a sub-curative treatment (T1) with LD-PIP, a curative treatment (T2) had to be administered before day 21 post challenge. The median period between T1 and T2 was 9.1 (range of 7.7–11.7), 10.0 (range of 9.2–10.2), 4.7 (range of 2–10.7), and 2.5 (range of 1.5–5.0) days for study arms LD-SP/SP, LD-SP/PIP, LD-PIP/PIP, and LD-PIP/SP, respectively. In participants receiving a subcurative LD-SP as T1, no recrudescent infection occurred and T2 was initiated on day 21 per protocol. One participant from treatment arm LD-PIP/PIP developed asexual recrudescence after T2, and received end treatment with atovaquone/proguanil on day 36. The remaining participants did not develop recrudescent infections after T2, and were treated with atovaquone/proguanil on day 42 as per protocol.

All participants also developed gametocytemia as determined by Pfs25 qRT-PCR (*Figure 2*; *Figure 3A*; *Figure 2—figure supplement 1*). Gametocytes were first detected 8.5–12 days after the initial peak of asexual parasites with no statistically significant difference in time to gametocyte appearance between study arms (p=0.26) (*Table 2*). The median peak density of gametocytes was 83 gametocytes/mL (range 11–1285) when all study participants were considered. Peak gametocyte

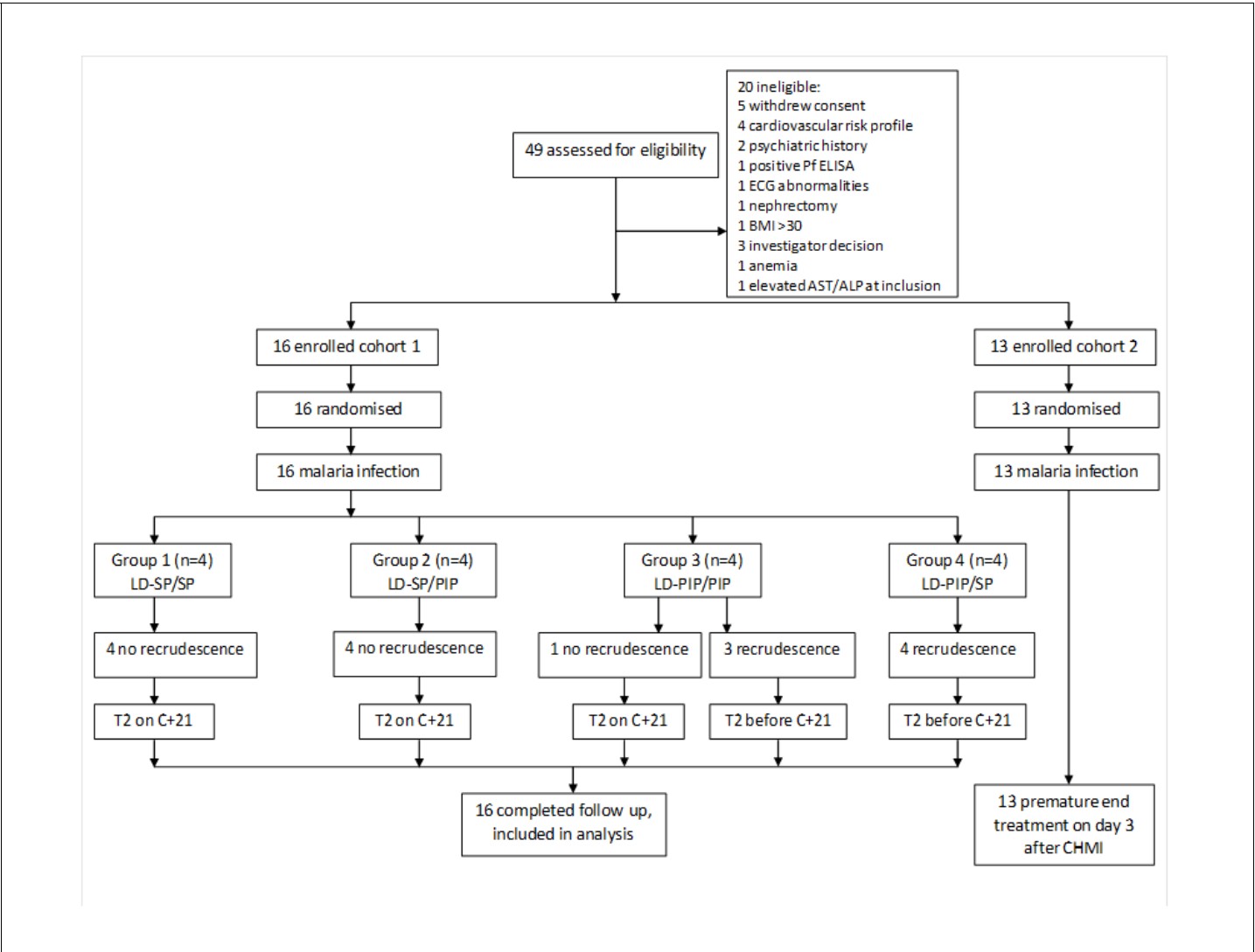

**Figure 1.** Trial profile. ECG = electrocardiography, BMI = body mass index, AST = aspartate aminotransferase, ALP = alkaline phosphatase

densities were higher in the study arm randomised to LD-PIP/SP, with a median of 627 gametocytes/mL (range of 199–1285), compared to 38 gametocytes/mL (range of 11–368), 30 gametocytes/mL (range of 13–101), 83 gametocytes/mL (range of 46–99), for arms LD-SP/SP, LD-SP/PIP, and LD-PIP/PIP, respectively (*Figure 2*; *Figure 2—figure supplement 1*; *Table 2*).

Thirteen (81%, 13/16) participants showed gametocytes on at least 5 consecutive days. The mean number of consecutive gametocyte-positive days was 24.5 (range of 17–25) for the LD-PIP/SP arm and was higher than for other arms (*Table 2*; *Figure 2*). Using multi-level logistic regression (random effect for within-group variation), we estimated that the average proportion of days that individuals tested positive for gametocytes was 27.4% (LD-SP/SP), 35.9% (LD-SP/PIP), 51.4% (LD-PIP/PIP), and 48.3% (LD-PIP/SP) (*Table 2*). The LD-PIP/PIP and LD-PIP/SP arms (i.e. those receiving 'low dose PIP') each had significantly higher average proportions of gametocyte-positive days than both arms LD-SP/SP and LD-SP/PIP (posterior probability 90.8% and 86.1%, respectively; 81.1% joint probability of arms LD-PIP/PIP and LD-PIP/SP both being higher than both LD-SP/SP and LD-SP/PIP). Furthermore, the area under the curve (AUC) for gametocyte density showed a statistically significant difference between arms (p=0.04). The LD-PIP/SP arm had a significantly higher gametocyte load (area under the curve) than each of the other three treatment arms (94.4% posterior probability of being the highest; *Figure 3B*). After correction for the asexual AUC, the probabilities of the gametocyte AUC in the LD-PIP/SP arm being higher than the other three decreased to 97.2%, 96.3%, and 96.2%

**Table 1.** Baseline characteristics of the participants included in analysis.

|  |  | LD-SP/SP | LD-SP/PIP | LD-PIP/PIP | LD-PIP/SP |
|---|---|---|---|---|---|
| No. subjects |  | n = 4 | n = 4 | n = 4 | n = 4 |
| Treatment 1 (T1) |  | Sulfadoxine-pyrimethamine 500 mg/25 mg | Sulfadoxine-pyrimethamine 500 mg/25 mg | Piperaquine 480 mg | Piperaquine 480 mg |
| Treatment 2 (T2) |  | Sulfadoxine-pyrimethamine 1000 mg/50 mg | Piperaquine 960 mg | Piperaquine 960 mg | Sulfadoxine-pyrimethamine 1000 mg/50 mg |
| Sex |  |  |  |  |  |
| Male | n (%) | 2 (50%) | 0 (0%) | 1 (25%) | 1 (25%) |
| Female | n (%) | 2 (50%) | 4 (100%) | 3 (75%) | 3 (75%) |
| Age | Mean (range) | 24.5 (21–29) | 24 (21–28) | 21.5 (20–24) | 22.5 (20–27) |
| BMI ($kg/m^2$) | Mean (range) | 21 (18–23) | 22 (19–25) | 24.5 (21–27) | 26.5 (24–29) |

The online version of this article includes the following source data for Table 1:

Source data 1. Source data for *Table 1*.

(from 99,1%, 98.9%, and 95.4%), and the probability of LD-PIP/SP being higher than all other study arms decreased to 94.0%.

Both female and male gametocytes were detected in 14/16 (88%) participants (*Figure 4*; *Figure 4—figure supplement 1*). Gametocyte sex-ratio's and circulation times have to be interpreted with caution since they rely on two separate qRT-PCR assays with differences in assay sensitivity (*Figure 5*; *Supplementary file 2*, *3*). On average 2.5 times as many female gametocytes were observed compared to male gametocytes per measured time-point (*Figure 4*; mean ratio 2.5 (SD = 2.5)). Combining all treatment arms, the best estimate of gametocyte half-life was 5.1 days (95% CI 4.1–6.1) for female gametocytes and 2.7 days (95% CI 1.5–3.9) for male gametocytes (*Figure 4—figure supplement 2*).

Gametocytes are produced from their asexual progenitors, and hence asexual parasite kinetics and gametocyte kinetics are related. The AUC of asexual parasitemia was statistically significantly associated with the AUC of gametocytemia ($r^2$ = 0.31, p=0.026), as shown in *Figure 3C*. The mean time-window between the first asexual parasites and the first appearance of gametocytes was 10.6 (SD = 0.65) days, see *Table 2*. Membrane feeding experiments were performed as an exploratory objective, and confirmed infectivity of gametocytes in three mosquitoes from three study arms on days 25 (LD-PIP/SP and LD-SP/SP arms) and 31 (LD-SP/PIP arm) post-infection. Mean gametocyte densities at those time-points were 106 gametocytes/mL (SD = 175), and 28 gametocytes/mL (SD = 47), respectively. Expressed as a proportion of all examined mosquitoes, 0.0002% (3/14400) of mosquitoes became infected in these exploratory assessments. Possible and probable related adverse events after challenge infection are shown in *Figure 6* and *Table 3*. The most frequently reported adverse events were fatigue, malaise, headache, fever, nausea, and chills. Grade three adverse events were reported in 14/16 (88%) participants, and were predominated by headache (n = 8), chills (n = 6), and nausea (n = 5). All possible and probable related adverse events resolved by the end of study. No serious adverse events occurred. The median number of adverse events was 20.5 per individual; the median number of adverse events with a grade three severity score was 1.5 per individual. There was no evidence for a difference between study arms in the occurrence of adverse events (p=0.49) or grade three adverse events (p=0.28).

Laboratory abnormalities during the study are shown in *Table 4*. Most prevalent abnormalities were elevated transaminases (ALT/AST) (n = 16), decreased lymphocytes (n = 15), decreased neutrophils (n = 13), and decreased platelets (n = 12). The only grade three laboratory abnormalities were elevated ALT (n = 8), and elevated AST (n = 7). 16/16 (100%) volunteers showed mild to severe ALT/AST elevations. 5/16 (31%) mild (grade 1); 3/16 (19%) moderate (grade 2), and 8/16 (50%) severe (grade 3) (up to 25 x ULN) ALT/AST elevations. These derangements were transient, and returned to baseline values within the normal range before the end of the study. A detailed overview of these liver function test derangements can be found in the supporting information (*Figure 6—figure*

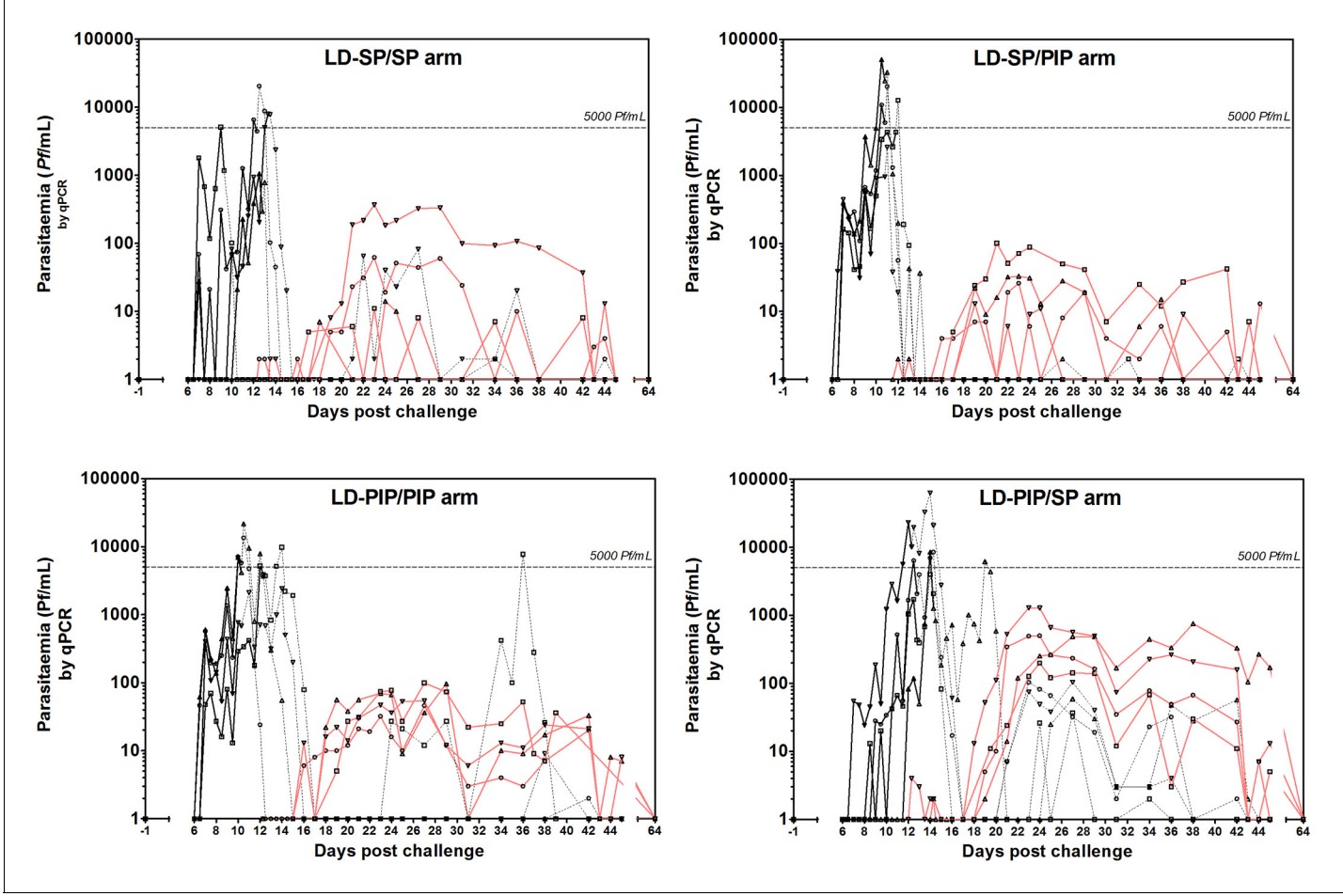

**Figure 2.** Asexual parasitemia and gametocytemia. Black line represents 18S qPCR asexual parasitemia. Black dotted-line represents 18S qPCR after treatment 1. Red line represents Pfs25 qRT-PCR gametocytemia.

The online version of this article includes the following figure supplement(s) for figure 2:

**Figure supplement 1.** Asexual parasitemia and gametocytemia per study participant.

*supplement 1*). These unexpected safety findings were reported to the Safety Monitoring Committee (SMC) and CCMO, and thoroughly reviewed.

## Discussion

Here, we present a CHMI model to induce mature gametocytes after mosquito bite infection in malaria-naïve study participants. The timing of the first appearance of gametocytes suggests that a fraction of the first wave of asexual parasites commit to the production of male and female gametocytes. With the use of antimalarial drugs that attenuates asexual stage infections but leave (developing) gametocytes unaffected, we determined biologically plausible half-lives of male and female gametocytes, and show preliminary evidence of the potential of this model to complete the lifecycle of malaria in mosquito feeding assays.

Malaria elimination efforts require a thorough understanding of the transmissibility of infections. Gametocyte commitment occurs for a fraction of asexual parasites under regulation of the transcription factor AP2-G with the entire progeny of a sexually committed schizont forming either male or female gametocytes (*Kafsack et al., 2014*). Our findings, based on novel sex-specific gametocyte qRT-PCR, confirm earlier work from malariatherapy studies where gametocytes were first detected by microscopy at 9–11 days after asexual parasites (*Ciuca et al., 1937*; *Shute and Maryon, 1951*). These data indicate very early gametocyte commitment and are in line with our earlier observations

**Table 2.** Treatment and parasitological data per study group.

| | | LD-SP/SP | LD-SP/PIP | LD-PIP/PIP | LD-PIP/SP |
|---|---|---|---|---|---|
| Time to T1 (days) | Median (range) | 13 (9.3–12.8) | 10.8 (0.8–11.8) | 10.3 (10.3–12.3) | 12.8 (12.3–14.3) |
| Time between T1-T2 (days) | Median (range) | 9.1 (7.7–11.7) | 10 (9.2–10.2) | 4.7 (2–10.7) | 2.5 (1.5–5.0) |
| Area under the curve (AUC)* | Median (range) | | | | |
| Asexual | | 6490 (1120–16337) | 13280 (2773–43777) | 14347 (5408–24898) | 12747 (4572–82973) |
| Sexual | | 280 (27–3640) | 271 (64–848) | 784 (316–1274) | 6624 (1515–10244) |
| Peak parasite density (Pf/mL) | Median (range) | 6467 (1050–20261) | 16376 (2590–50210) | 11603 (2408–21565) | 8491 (3976–63113) |
| Peak gametocyte density (gct/mL) | Median (range) | 38 (11–368) | 30 (13–101) | 83 (46–99) | 627 (199–1285) |
| Day of gametocyte detection after infection (days) | Mean (SD) | 18.3 (1.0) | 18.5 (1.0) | 17.3 (1.5) | 19.4 (1.3) |
| Time to gametocyte detection relative to first asexual parasites[†] (days) | Mean (SD) | 10.5 (1.3) | 11.5 (1.0) | 10.1 (1.3) | 10.1 (1.2) |
| Proportion of days gametocyte positive (%)[‡] | Mean (SD) | 27.4 (6.7) | 35.9 (7.6) | 51.4 (7.9) | 48.3 (8.1) |
| Duration gametocytemia[§] (days) | Median (range) | 7.5 (1–24) | 6 (2–14) | 17 (12–25) | 24.5 (17–25) |

*The area under the curve (AUC) represents the total parasite exposure over time (asexual- or sexual parasite load).

[†]Time to gametocyte detection is calculated as the day of the detection of gametocytes (≥5 gct/mL) minus the day of first peak asexual parsitaemia.

[‡]The proportion of gametocyte positive days is calculated as all days with ≥5 gct/mL by Pfs25-qRT-PCR divided by all days where Pfs25 qRT-PCR was performed.

[§]Maximum number of consecutive days of Pfs25 qRT-PCR measured gametocytemia ≥5 gct/mL.

that Pfs16 mRNA, the earliest gametocyte transcript, is detectable at the moment of peak parasitemia in CHMI models (*Schneider et al., 2004*). This timing is highly relevant for understanding gametocyte transmission biology. The circulation of mature gametocytes has not been reported in previous CHMI trials using curative regimens of chloroquine, artemether-lumefantrine, or atovaqoune-proguanil, and our data illustrate the differential impact of antimalarial drugs on developing gametocytes. Once treatment is initiated, gametocyte production ceases abruptly (in the case of artemisinins), remains unaffected, or may even be stimulated under drug pressure as suggested for sulfadoxine-pyrimethamine and piperaquine (*Bousema and Drakeley, 2011*; *Butcher, 1997*; *Adjalley et al., 2011*). In our study, we aimed for a protracted low density of asexual parasitemia demonstrating that early abrogation of asexual infections by both sulfadoxine-pyrimethamine and piperaquine permits successful mature gametocyte development. SP has long been associated with a rapid appearance of gametocytes that is too early to be explained by de novo gametocyte production upon drug pressure and has thus been hypothesized to reflect an efflux of sequestered gametocytes upon treatment (*Butcher, 1997*). Evidence for the permissiveness of piperaquine to (developing) gametocytes is more recent (*Pasay et al., 2016*; *Farid et al., 2017*; *Adjalley et al., 2011*). In the current study, group sizes are limited and comparisons between treatment arms have to be interpreted with caution. CHMI studies are logistically challenging and the number of volunteers that can be monitored to ensure participant safety is an important consideration when defining the study size. Our sample size calculation was based on the optimistic assumption that the vast majority of volunteers would develop mature gametocytes; an assumption that was supported by the current data. With our limited study size, our findings indicate that none of the study drugs prevented the appearance of gametocytes after treatment, thereby suggesting limited or no effect of PIP and SP on developing or mature gametocytes (*Bolscher et al., 2015*). We hypothesized that slow acting drugs may promote the development of gametocytes (*Méndez et al., 2002*), potentially via microvesicles that are derived from infected erythrocytes (*Nilsson et al., 2015*) and differences between drug regimens in the rate at which asexual parasites are cleared upon T1 and T2 would

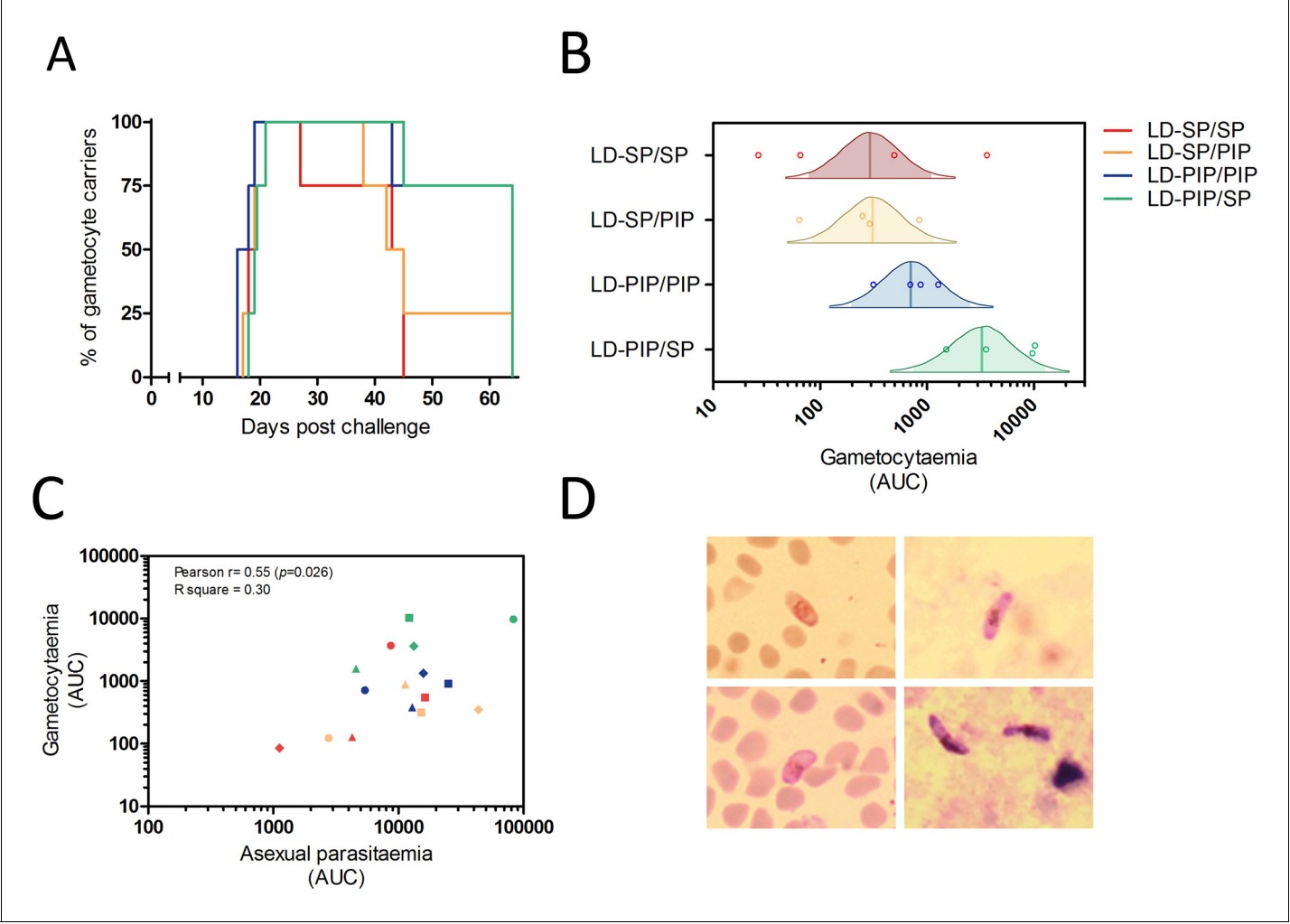

**Figure 3.** Gametocyte kinetics between study arms. (A) Percentage gametocyte carriers between study arms (B) Estimated mean area under the curve for concentration of gametocytes per arm (Bayesian framework). The shaded area of each density curve represents the middle 95% percentiles (i.e. 2.5th to 97.5th percentiles) of the estimated mean AUC for a study arm; the density curve itself spans the middle 99% percentiles of the posterior; the posterior mean is indicated by the vertical solid line within each density plot. (C) Association of area under the curves of asexual parasitemia and gametocytemia. The different plotting shapes are the individual participants per group. (D) Thin- and thick- blood smears of concentrated gametocytes after magnetic cell sorting of blood samples from two individuals from LD-PIP/SP arm.

result in different gametocyte dynamics. Although our findings indicate highest gametocyte concentrations in the LD-PIP/SP arm, more observations and thus additional studies are needed to allow the construction of a model that allows a quantification of gametocyte commitment at different time-points during the study (e.g. prior to T1, during the phase of parasite recrudescence and following T2). One hypothesis would be similar gametocyte commitment in all arms after T1 but a more rapid release of gametocytes that accumulated in the bone marrow between T1 and T2.

We present the novel evidence that both male and female gametocytes appear early, upon infection. Our findings suggest an earlier appearance of female gametocytes (18.8 days (SD 1.8) compared to male gametocytes 20.3 days (SD 1. 2)) and a longer circulation time of female gametocytes that is in line with previous estimates from naturally infected individuals (*Bousema et al., 2010*; *Ciuca et al., 1937*). Whilst both male and female gametocytes are consistently detected at densities of 0.1 gametocyte/μL (*Stone et al., 2017*), the highly abundant Pfs25 mRNA makes the female gametocyte qRT-PCR more sensitive than the male PfMGET qRT-PCR. Differences in gametocyte dynamics between male and female gametocytes should therefore be interpreted with caution. Gametocyte densities remained below the threshold of detection by microscopy throughout the

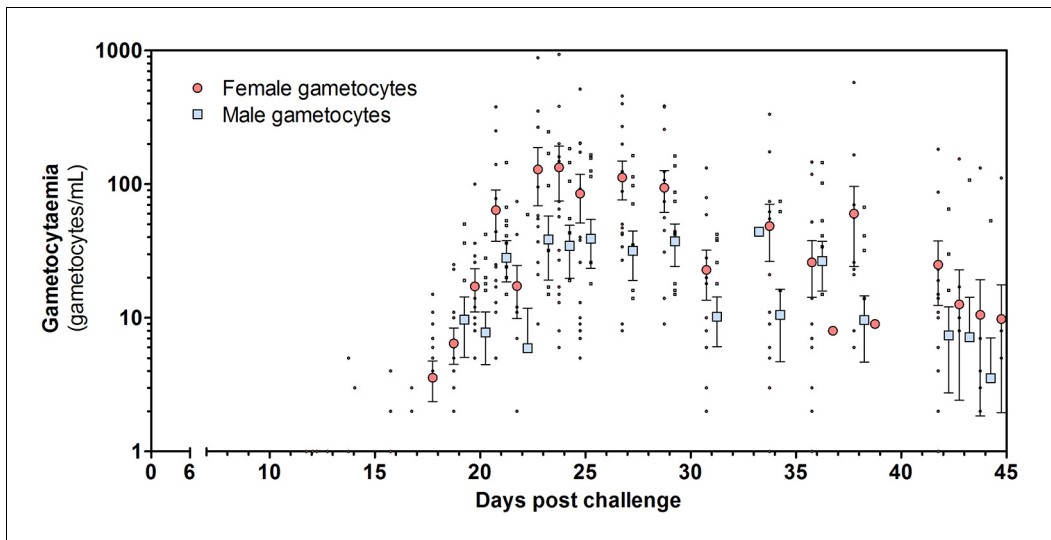

**Figure 4.** Total female and male gametocyte density of all participants. Dots represent individual gametocyte data. Circles and squares represent mean and error (SEM) of gametocytes per timepoint.

The online version of this article includes the following figure supplement(s) for figure 4:

**Figure supplement 1.** Female and male gametocytes per study arm.

**Figure supplement 2.** Female and male gametocyte clearance dynamics per participant included in analysis.

study period and were strongly associated with the preceding densities of the asexual progenitors. Participants in the LD-PIP/SP study arm showed the highest gametocytes densities, and a mean female/male sex ratio of 4.1 (SD = 5.1), in line with gametocyte sex-ratios in natural infections (~3 to 5 females to one male) (*Ciuca et al., 1937*; *Delves et al., 2013*). We confirmed the infectivity of gametocytes in three mosquitoes from three study arms. The very low rate of infected mosquito corroborates observations from naturally acquired infections where mosquito infection becomes highly unlikely below 1000–10,000 gametocytes/mL (*Gonçalves et al., 2016*). The sporadic mosquito infections thus demonstrate that mature gametocytes in sex-ratios supportive of mosquito infections can

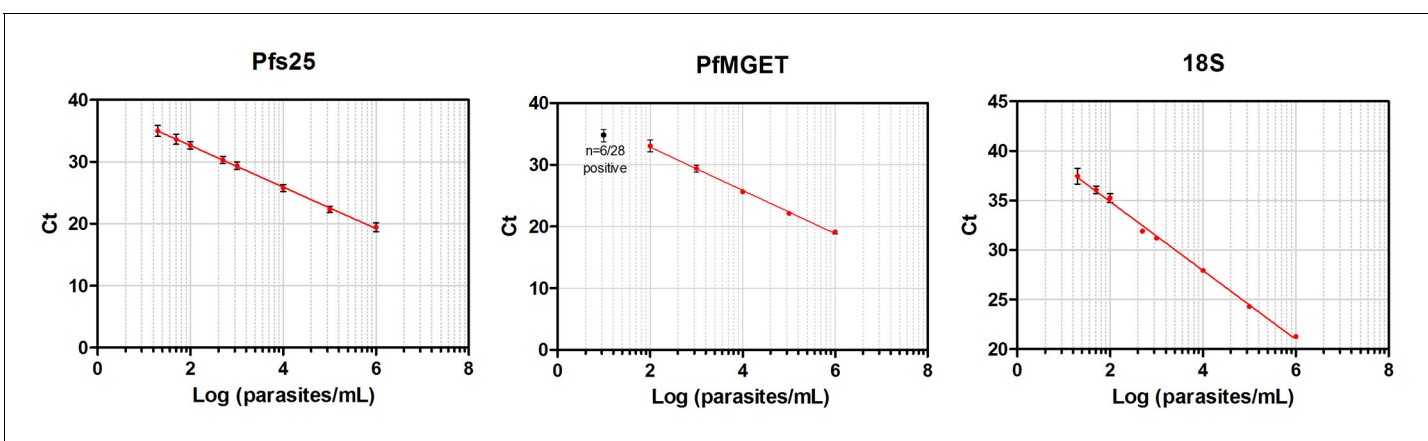

**Figure 5.** Standard curves of qRT-PCR and qPCR. Standard curves (Mean, SD) obtained using 10-fold dilutions of cultured gametocytes. The highest concentration was enumerated by two independent expert microscopists. The mean and standard deviation of 54, 28, 72 replicates of the standard curve during the study was determined for the Pfs 25-, PfMGET, and 18S target genes, respectively. For PfMGET, six points starting from $10^6$ pure male gametocytes/mL were measured. $10^1$ was positive in 6/28 replicates (black dot).

The online version of this article includes the following figure supplement(s) for figure 5:

**Figure supplement 1.** Standard curves of Pfs25 qRT-PCR – low-density trendlines.

**Figure supplement 2.** Correlation of duplo Pfs25 qRT-PCR measurements in all study samples.

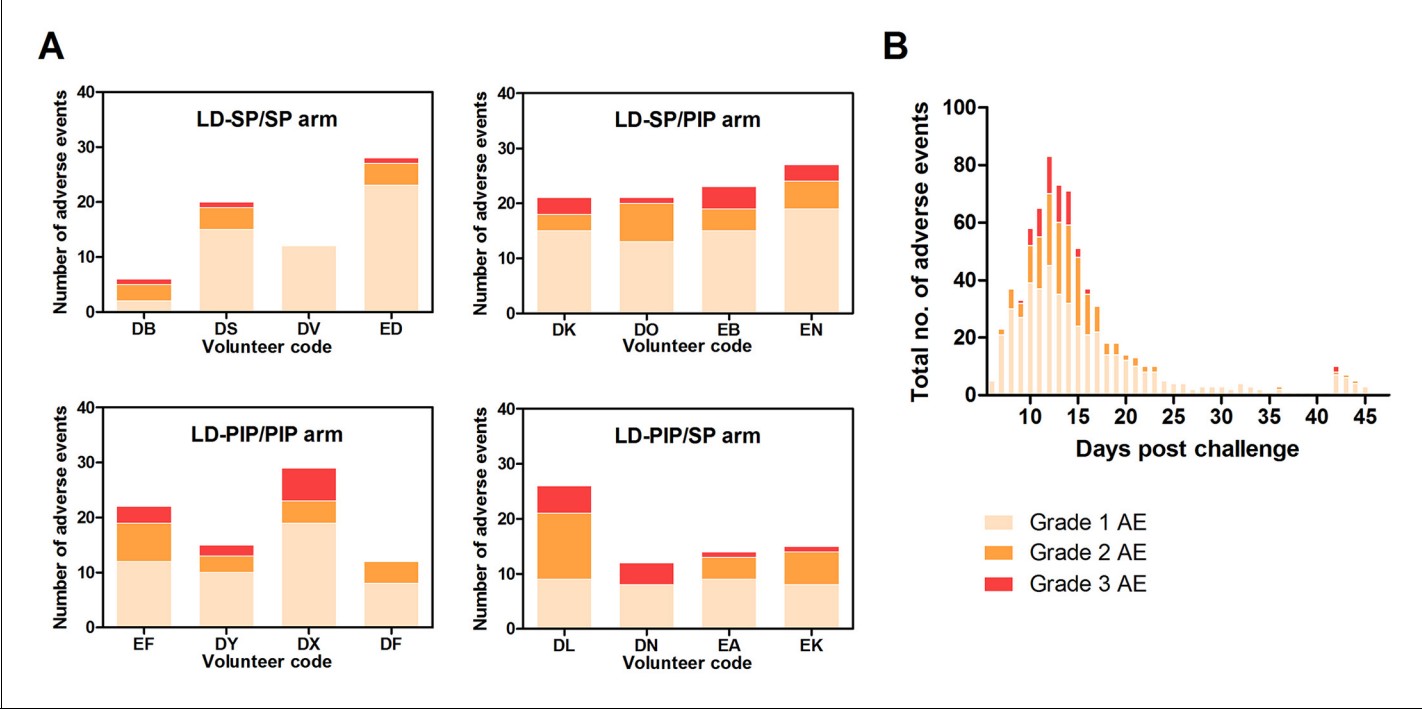

**Figure 6.** Adverse events. (A) Adverse events per study arm (B) Total no. of adverse events and time course.
The online version of this article includes the following figure supplement(s) for figure 6:

**Figure supplement 1.** Liver function test derangements.

be achieved in CHMI transmission models. Studies on the evaluation of TBIs will need a further optimized protocol aimed to achieve higher gametocyte densities by increasing duration and load of the asexual parasite burden. For the evaluation of gametocytocidal interventions in the CHMI transmission model, gametocyte densities should be sufficiently high to quantify an intervention-associated reduction in gametocyte appearance or gametocyte half-life. For the evaluation of interventions that reduce the transmissibility of gametocytes, higher mosquito infections should be achieved at proportions that allow the detection of meaningful reductions in mosquito infection rates in experimental arms. Low infectivity in membrane feeding assays may be overcome by achievement of higher gametocyte densities in the model, and the use of gametocyte concentration methods (*Reuling et al., 2017*), or by direct skin feeding assays (*Bousema et al., 2012*).

In line with recent findings, we observed recrudescent infections in 7/8 participants treated with LD-PIP (*Pasay et al., 2016*). Recrudescent infections were not observed in arms that first received LD-SP, suggesting that this dose, although 1/3 of the standard curative dose of sulfadoxine-pyrimethamine, is curative at asexual parasite densities observed in our participants. It has been hypothesized that the prolonged parasitemia under drug pressure increases gametocyte commitment (*WWARN Gametocyte Study Group, 2016*). The duration of parasite multiplication between T1 and T2 was relatively short in this study (2–5 days) for subjects with recrudescent infections, and the contribution of drug pressure may thus have been limited. The current findings suggest that further lowering the SP dose may be considered to prolong asexual parasite exposure.

The liver enzyme elevations found in our study led to a structured risk analysis, and review by independent experts. Transient, asymptomatic liver function test (LFT) derangements have been reported in volunteers in previous CHMI studies, and are likely to be related to the asexual stage parasitemia, and subsequent treatment.

Detailed studies on gametocyte biology and dynamics, and the early development of novel drugs and vaccines that target malaria transmission (TBIs) are currently restricted to in vitro assays, such as drug sensitivity assays, and standard membrane feeding assays (SMFA) (*Bousema and Drakeley, 2011*; *Wells et al., 2009*). Recently, a humanized mice model has been developed to investigate *P.*

**Table 3.** List of adverse events possibly or probably related to the trial.

| Adverse events | Total | LD-SP/SP | | | LD-SP/PIP | | | LD-PIP/PIP | | | LD-PIP/SP | | |
|---|---|---|---|---|---|---|---|---|---|---|---|---|---|
| | Number of subjects | Number of subjects | Number of episodes | Mean duration in days (SD) | Number of subjects | Number of episodes | Mean duration in days (SD) | Number of subjects | Number of episodes | Mean duration in days (SD) | Number of subjects | Number of episodes | Mean duration in days (SD) |
| Fatigue, Malaise | 16 | 4 | 10 | 3.6 (4.5) | 4 | 15 | 2.0 (3.0) | 4 | 10 | 2.9 (1.1) | 4 | 6 | 6.8 (8.1) |
| Headache | 15 | 3 | 12 | 1.0 (1.5) | 4 | 25 | 1.2 (1.2) | 4 | 17 | 1.3 (1.2) | 4 | 21 | 1.6 (1.4)) |
| Fever | 15 | 4 | 9 | 0.4 (0.4) | 4 | 10 | 0.3 (0.4) | 3 | 11 | 0.4 (0.3) | 4 | 13 | 0.7 (0.4) |
| Nausea | 14 | 4 | 12 | 0.6 (0.8) | 4 | 15 | 1.1 (1.6) | 3 | 8 | 1.2 (1.5) | 3 | 10 | 0.7 (1.0) |
| Chills | 14 | 3 | 4 | 1.7 (1.0)) | 3 | 5 | 1.7 (2.0) | 4 | 10 | 1.2 (1.3) | 4 | 6 | 0.9 (1.1) |
| Myalgia | 11 | 3 | 5 | 3.2 (3.3) | 3 | 9 | 2.1 (1.9) | 3 | 5 | 1.2 (1.0) | 2 | 3 | 2.2 (2.6) |
| Abdominal pain | 10 | 2 | 5 | 0.3 (0.2) | 3 | 3 | 0.6 (0.9) | 2 | 8 | 1.1 (1.3) | 3 | 3 | 1.6 (2.4) |
| Pruritis | 6 | 2 | 3 | 0.6 (0.8) | 2 | 2 | 3.3 (0.5) | 1 | 2 | 0.3 (0.4) | 1 | 1 | 3.6 |
| Athralgia | 5 | 1 | 1 | 2.2 | 2 | 4 | 1.5 (1.8) | 0 | - | - | 2 | 2 | 5.1 (3.6) |
| Diarrhoea | 5 | 1 | 1 | 0.8 | 1 | 1 | 0.1 | 2 | 2 | 1.7 (2.1) | 1 | 1 | 4 |
| Diziness | 3 | 1 | 1 | 0.1 | 0 | - | - | 2 | 5 | 0.5 (0.7) | 0 | - | - |
| Reflux | 2 | 0 | - | - | 2 | 2 | 2.9 (1.8) | 0 | - | - | 0 | - | - |
| Pyrosis | 1 | 0 | - | - | 0 | - | - | 0 | - | - | 1 | 1 | 8.6 |
| Aspecific chest pain | 1 | 1 | 2 | 0.0 (0.0) | 0 | - | - | 0 | - | - | 0 | - | - |
| Syncope | 1 | 0 | - | - | 1 | 1 | 0.0 | 0 | - | - | 0 | - | - |
| Mouth ulcera | 1 | 1 | 1 | 10.0 | 0 | - | - | 0 | - | - | 0 | - | - |
| **Grade 3 adverse events** | | | | | | | | | | | | | |
| Total | 14 | 3 | | | 4 | | | 3 | | | 4 | | |
| Headache | 8 | 0 | - | - | 2 | 2 | 0.3 (0.2) | 2 | 2 | 0.6 (0.1) | 4 | 4 | 1.1 (1.3) |
| Chills | 6 | 1 | 1 | 0.9 | 2 | 2 | 1.7 (2.0) | 2 | 2 | 0.3 (0.3) | 1 | 1 | 2.2 |
| Nausea | 5 | 1 | 1 | 0.1 | 2 | 3 | 0.3 (0.6) | 1 | 1 | 0.7 | 1 | 1 | 0 |
| Fever | 4 | 0 | - | - | 0 | - | - | 2 | 5 | 0.5 (0.4) | 2 | 5 | 0.7 (0.5) |
| Fatigue, malaise | 4 | 0 | - | - | 3 | 4 | 0.8 (0.4) | 1 | 1 | 2 | 0 | - | - |
| Abdominal pain | 1 | 1 | 1 | 0.5 | 0 | - | - | 0 | - | - | 0 | - | - |

*falciparum* sexual commitment that could, therefore, bridge in vitro assays to in vivo animal studies that take into account drug metabolism and gametocyte sequestration (*Duffier et al., 2016*). Also, an experimental *Plasmodium vivax* transmission model in human has been reported (*Griffin et al., 2016*). However, mechanisms underlying P. *falciparum* gametocytogenesis and dynamics have never been addressed in a controlled clean system in humans.

Here, we present a novel CHMI transmission model for *P. falciparum* that can be used to study gametocyte biology and dynamics providing novel insights and tools in malaria transmission and elimination efforts. The dynamics of gametocyte commitment, maturation, sex ratio, and sequestration found in our model reflect parasite dynamics found in naturally acquired infections, although parasite densities are much lower than in many endemic settings. This model can be used to evaluate the effect of drugs and vaccines on gametocyte dynamics and sex ratios. With its current performance, the CHMI transmission model may allow testing of vaccination strategies that reduce the

**Table 4.** Laboratory abnormalities per study arm.

| | LD-SP/SP | | | LD-SP/PIP | | | LD-PIP/PIP | | | LD-PIP/SP | | |
|---|---|---|---|---|---|---|---|---|---|---|---|---|
| | N (% of total) of grade 1 | N (% of total) of grade 2 | N (% of total) of grade 3 | N (% of total) of grade 1 | N (% of total) of grade 2 | N (% of total) of grade 3 | N (% of total) of grade 1 | N (% of total) of grade 2 | N (% of total) of grade 3 | N (% of total) of grade 1 | N (% of total) of grade 2 | N (% of total) of grade 3 |
| Any lab. abnormality | 15 (14) | 7 (7) | 2 (2) | 13 (12) | 10 (9 | 3 (3) | 16 (15) | 9 (8) | 2 (2) | 13 (12) | 8 (8) | 8 (8) |
| Decreased hemoglobin | 0 | 0 | 0 | 1 (14) | 2 (29) | 0 | 1 (14) | 1 (14) | 0 | 1 (14) | 1 (14) | 0 |
| Decreased WBC | 1 (8) | 3 (23) | 0 | 1 (8) | 2 (15) | 0 | 1 (8) | 2 (15) | 0 | 1 (8) | 2 (15) | 0 |
| Decreased neutrophils | 3 (23) | 1 (8) | 0 | 2 (15) | 0 | 0 | 3 (23) | 1 (8) | 0 | 3 (23) | 0 | 0 |
| Decreased lymphocytes | 3 (20) | 1 (7) | 0 | 1 (7) | 3 (20) | 0 | 3 (20) | 1 (7) | 0 | 1 (7) | 2 (13) | 0 |
| Decreased platelets | 3 (25) | 0 | 0 | 2 (17) | 0 | 0 | 4 (33) | 0 | 0 | 1 (8) | 2 (17) | 0 |
| Elevated ALT | 2 (13) | 1 (6) | 1 (6) | 2 (13) | 0 | 2 (13) | 1 (6) | 2 (13) | 1 (6) | 0 | 0 | 4 (25) |
| Elevated AST | 1 (7) | 1 (7) | 1 (7) | 2 (13) | 1 (7) | 1 (7) | 1 (7) | 2 (13) | 1 (7) | 0 | 0 | 4 (27) |
| Elevated yGT | 1 (11) | 0 | 0 | 1 (11) | 1 (11) | 0 | 2 (22) | 0 | 0 | 3 (33) | 1 (11) | 0 |
| Elevated ALP | 0 | 0 | 0 | 0 | 1 (33) | 0 | 0 | 0 | 0 | 2 (67) | 0 | 0 |
| Elevated total bilirubin | 1 (50) | 0 | 0 | 0 | 0 | 0 | 0 | 0 | 0 | 1*(50) | 0 | 0 |
| Elevated creatinine | 0 | 0 | 0 | 1 (100) | 0 | 0 | 0 | 0 | 0 | 0 | 0 | 0 |
| Elevated BUN | 0 | 0 | 0 | 0 | 0 | 0 | 0 | 0 | 0 | 0 | 0 | 0 |

Number of subjects with the highest grade reported for a laboratory abnormality. Grading based on WHO toxicity grading scale. No grade four abnormalities were reported. Lymphocytes ($10^9$/l) were graded based on grade 1: 0.9–0.6; grade 2: 0.3–0.5; grade 3:<0.3.

Liver function tests were graded based on grade 1: 1.1.–2.5X ULN, grade 2: 2.6–5.0x ULN, grade 3:>5.0X ULN. WBC, white blood count; ALT, alanine aminotransferase; AST, aspartate aminotransferase; yGT, glutamyl transpeptidase; ALP, alkaline phosphatase;

See *Figure 6—figure supplement 1* for a detailed overview of liver function test abnormalities.

BUN, blood urea nitrogen. T1, treatment 1; T2, treatment 2.*Subject showed elevated total bilirubin at baseline.

production of gametocytes from their asexual progenitors or accelerate their clearance from the blood stream (*Stone et al., 2016*), and the testing of gametocytocidal drugs (*White, 2013*). To allow testing of sterilizing effect of drugs on circulating gametocytes (*White et al., 2014*) or the effect of antibodies that interfere with gametocyte fertilisation inside the mosquito gut (*Stone et al., 2016*), the model needs to be optimized to achieve considerably higher mosquito infection rates. The current work lays the foundation for fulfilling the critical unmet need to evaluate transmission-blocking interventions against *falciparum* malaria for downstream selection and clinical development.

## Materials and methods

### Study design

This single centre, open-label randomised trial was conducted at the Radboud university medical center (Radboudumc), Nijmegen, the Netherlands. Healthy malaria-naive male and female participants aged 18–35 years were recruited from June until November 2016. Screening included physical examination, electrocardiography (ECG), hematology and biochemistry parameters and serology for human immunodeficiency virus (HIV), hepatitis B and C, and asexual stages of *P. falciparum*. Informed consent was provided by all participants at screening visit. The central committee for research involving human subjects (CCMO), and the Western Institutional Review Board (WIRB) approved the protocol for this study (NL56659.091.16). The trial was conducted according to the

principles outlined in the Declaration of Helsinki and Good Clinical Practice standards, and registered at ClinicalTrials.gov, identifier NCT02836002 (*Supplementary file 4*; Reporting Standard 1).

## Randomisation

A total of 16 participants were included in the analysis of this study. After inclusion, study participants were randomly allocated to one of the four different treatment arms (n = 4 per group) with low-dose (LD) of either piperaquine (PIP) or sulfadoxine-pyrimethamine (SP), followed by curative regimen of piperaquine or sulfadoxine-pyrimethamine upon recrudescence; (i) LD-SP/SP, (ii) LD-SP/PIP, (iii) LD-PIP/SP, or (iv) LD-PIP/SP. Randomisation was done by a computer-generated random number table (Microsoft Excel 2007, Redmond, WA).

## Procedures

All study participants were subjected to a standard CHMI with five female *Anopheles stephensi* mosquitoes infected with the *P. falciparum* strain 3D7 (*Sauerwein et al., 2011*; *Cheng et al., 1997*). *P. falciparum* 3D7 asexual and sexual blood stages were cultured in a semi-automated culture system and used to infect mosquitoes by standard membrane feeding as described previously (*Ponnudurai et al., 1986*; *Ponnudurai et al., 1989*). The 3D7 lineage that was used in the current study is based on a 3D7 bank described in detail in *Cheng et al. (1997)*. To examine molecular markers of drug resistance, we used available Illumina whole genome sequencing data (https://www.ebi.ac.uk/ena/data/view/PRJEB12838); aligning reads to the *P. falciparum* reference genome v3 (plasmoDB) with bowtie2 (sourceforge) and obtaining consensus sequences for dhps and dhfr genes with samtools. No mutations were identified in the dhfr gene; the only detected mutation was dhps A437G which, by itself, is not associated with sulfadoxine-pyrimethamine resistance (*Staedke et al., 2004*). Plasmepsin II/III duplication events are associated with piperaquine resistance (*Witkowski et al., 2017*) but were not observed although the sequence similarities with neighboring genes Plasmepsin I and IV suggest that unambiguous quantification may require more specific gene targeting. Importantly, piperaquine sensitivity of our 3D7 lineage was previously confirmed by in vivo experiments (*Pasay et al., 2016*). We conclude that the lineage used was sensitive to both sulfadoxine-pyrimethamine and piperaquine.

Participants were monitored twice daily on an outpatient basis from day 6 after exposure to infected mosquitoes until malaria parasites were detected at a density of ≥5000 parasites per milliliter (*Pf*/mL) by qPCR or a positive thick blood smear, upon which they were treated with a subcurative dose of 500 mg/25 mg sulfadoxine-pyrimethamine (Roche, Boulogne-billancourt, FR) or 480 mg of piperaquine phosphate (PCI Pharma Services, Tredegar, UK). After the first treatment (T1), participants continued to visit the study center twice daily for another 4 days to monitor the initial clearance of parasitemia by qPCR, after which they were monitored once a day for recrudescence. On day 21 or upon parasite density reaching ≥1500 *Pf*/mL, participants received a second treatment (T2), consisting of 1000 mg/50 mg sulfadoxine-pyrimethamine or 960 mg of piperaquine phosphate. After the second treatment, participants were monitored daily for 3 days, then three times a week until final treatment with atovaquone/proguanil (Malarone) on day 42. Adverse events were recorded, and blood sampling was performed to monitor parasitemia and blood safety parameters. Symptoms of malaria were treated with acetaminophen up to 4000 mg daily, and nausea with metoclopramide up to 30 mg daily, if necessary.

Parasite density was determined by quantitative PCR (qPCR) targeting the multicopy 18S rRNA gene (*Hermsen et al., 2001*); samples collected in the morning were processed immediately, evening samples 12 hr later. Thick blood smears were taken during evening visits, double-read and considered positive if two or more parasites were detected in 0.5 μL (*Laurens et al., 2012*). The presence of gametocytes was monitored in samples from day 7.5 after challenge until end of study by quantitative reverse-transcriptase PCR (qRT-PCR) targeting female-specific Pfs25 mRNA and male specific PfMGET (Pf3D7_1469900) and using sex-specific trendlines (*Stone et al., 2017*; *Pett et al., 2016*). All samples with an estimated gametocyte density ≥5 gametocytes per mL (gametocytes/mL) were considered gametocyte positive. The duration of gametocyte carriage as an indicator of stable gametocyemia was defined as the maximum number of consecutive days with detectable gametocytemia above the threshold for detection. Direct Membrane Feedings Assays (DMFA) were performed as exploratory measures on days 21, 25 and 31 post-infection with ~300 mosquitoes per

feed per participant (total of ~14,400 mosquitoes) (*Bousema et al., 2013*; *Lensen et al., 1998*; *Ouédraogo AL et al., 2013*). Mosquito infection status was determined on day 12 by circumsporozoite (CSP) ELISA(*Stone et al., 2015*) followed by qPCR confirmation of mosquitoes where the OD exceeded the mean +3 standard deviations of control mosquitoes (*Graumans et al., 2017*).

Adverse events were recorded and graded by the research physician as mild (easily tolerated, grade 1), moderate (interfering with daily activity, grade 2) or severe (preventing daily activity, grade 3), and in the case of fever as mild (38.0–38.4°C), moderate (38.5–38.9,°C) or severe (≥39°C). Safety blood tests were performed daily, including full blood counts, LDH and highly sensitive troponin-T. Biochemistry tests including liver function test were assessed at screening, inclusion, 2 days after every treatment and at the end of study, and on additional days if considered relevant for clinical decision-making.

## Pfs25 and PfMGET RNA quantification

For the quantification of the *P. falciparum* Pfs25 transcript levels total NA was RQ1 DNaseI treated according to the manufacturer's protocol. 2 µL of DNaseI-treated material was run in a total volume of 25 µL of TaqMan RNA-to-Ct qRT-PCR reaction mixture (Applied Biosystems, Foster City, California). For the quantification of the *P. falciparum* male gametocyte enriched transcript (PfMGET), cDNA was synthesized from Total NA with the High Capacity cDNA Reverse Transcription Kit (Applied Biosystems). Samples were added in a 1: one ratio to the mastermix. 2 µL of cDNA was run in a total volume of 20 µL making use of the GoTaq qPCR Master Mix (Promega, Madison, Wisconsin). Male *P. falciparum* gametocytes were quantified using a standard curve of serially diluted StageV male gametocytes from the transgenic PfDynGFP/P47mCherry line (*Lasonder et al., 2016*). Detailed information on the validation and performance characteristics of the assays can be found in the supporting materials (*Figure 5*; *Supplementary file 2*, *3*; *Figure 5—figure supplement 1*, *2*).

## Study outcome

The primary study outcomes were the frequency and magnitude of adverse events, and the prevalence of gametocytes by Pfs25 qRT-PCR. The prevalence of gametocytes is the presence of female gametocytes as measured by qRT-PCR targeting female-specific Pfs25 mRNA at any of the twice daily measurements from day 6. Secondary outcomes were the peak density and time-point of peak density of male and female gametocytes, the AUC of gametocyte density, and assessment of the dynamics of gametocyte commitment, maturation and sex-ratio. The AUC of gametocyte density represents the total gametocyte exposure over time (gametocyte load). Assessment of gametocyte infectivity to *Anopheles stephensi* mosquitoes by DMFA was an exploratory study endpoint.

## Statistical analysis

The sample size was calculated based on preliminary data that > 95% of the participants would develop gametocytemia. Conservatively, we considered the approach unsuitable for gametocyte induction if <50% of individuals developed mature gametocytes. We, therefore, powered the trial to estimate a 90% confidence interval around the proportion of gametocytaemic individuals that excludes 50%. If eight individuals (allowing for one dropout per arm), and 6/7 or 7/7 of these individuals become gametocytaemic, we would be able to estimate this proportion with a lower limit of the 90% Wilson confidence interval ≥54.8% (the lower limit of the 95% confidence interval being 48.7%). Differences between study arms were assessed by comparing mean values using a one-way ANOVA or non-parametric equivalents.

To further identify which study arm(s) potentially deviated from others, we jointly estimated the differences between all four arms in a Bayesian framework (standard linear regression model, no mixed effects), using Hamiltonian Monte Carlo as implemented in the R package *rstanarm*, and using an uninformative (uniform) prior for the explained variation ($R^2$) (see R codes used in *Source code 1*) (*Team SD, 2016*). For discrete variables (e.g. the number of positive assays), the chi-squared test or Fisher's exact test was used (two-tailed). The total number of adverse events and total number of grade three adverse events were calculated per individual and compared by non-parametric Kruskal Wallis test.

A previously developed model was used to estimate gametocyte half-life for female and male gametocytes separately (*Bousema et al., 2010*). For this analysis, gametocyte observations were included from 12 days after the last detection of asexual parasites until the end of study. This was based on the gametocyte sequestration time of 10–12 days in this study, and the assumption that the number of newly released gametocytes would thus be minimal in this observation period. All model fittings were carried out using the PROC NLMIXED procedure in SAS (Version 9, SAS Institute Inc) and included no covariates other than time (see *Source code 2* for SAS code). The AUC was computed by GraphPad Prism 5 (USA) with the (X2-X1)*(Y1 +Y2)/2 formula (X = days post challenge; Y = gametocytes per mL ($\geq$5 gametocytes/mL)) as used repeatedly for each adjacent pair of points defining the curve; the total AUC was used.

## Acknowledgements

We would like to thank the staff from the Clinical Research Centre Nijmegen. We also thank Daphne Smit and Annemieke Jansens for their assistance and project support during the trial. We thank the following individuals for their assistance during the trial: Laura Pelser, Jolanda Klaassen, Astrid Pouwelsen, and Jacqueline Kuhnen for the mosquito infection and dissection work, Foekje Stelma, and all the thick smear microscopists for reading many smears. We acknowledge Gheorghe Pop for his help in evaluation of electrocardiograms. This trial was supported mainly with funds from Path Malaria Vaccine Initiative (MVI). TB is supported by a fellowship from the European Research Council (ERC-2014-StG 639776). Finally and foremost, we would like to thank the study volunteers who participated in this trial. The funder, PATH's Malaria Vaccine Initiative, was involved in the study design, analysis and interpretation of the data, the preparation of the report, but not data collection. IJR, LS, TB, and RWS had full access to all study data with final responsibility for the decision to submit the report for publication.

## Additional information

### Funding

| Funder | Grant reference number | Author |
|---|---|---|
| PATH | Malaria Vaccine Initiative | Isaie J Reuling<br>Lisanne A van de Schans<br>Kjerstin Lanke<br>Wouter Graumans<br>Geert-Jan van Gemert<br>Karina Teelen<br>Rianne Siebelink-Stoter<br>Marga van de Vegte-Bolmer<br>Quirijn de Mast<br>André J van der Ven<br>Karen Ivinson<br>Cornelus C Hermsen<br>Katharine A Collins<br>Christian F Ockenhouse<br>James McCarthy<br>Robert W Sauerwein<br>Teun Bousema |
| H2020 European Research Council | Fellowship | Teun Bousema |
| European Research Council | ERC-2014-StG 639776 | Teun Bousema |

The funder, PATH's Malaria Vaccine Initiative, was involved in the study design, analysis and interpretation of the data, the preparation of the report, but not data collection. IJR, LS, TB, and RWS had full access to all study data with final responsibility for the decision to submit the report for publication.

## Author contributions

Isaie J Reuling, Conceptualization, Data curation, Formal analysis, Validation, Investigation, Visualization, Methodology, Writing—original draft, Project administration, Writing—review and editing; Lisanne A van de Schans, Formal analysis, Investigation, Visualization, Writing—original draft, Writing—review and editing; Luc E Coffeng, Sake de Vlas, John Bradley, Formal analysis, Writing—review and editing; Kjerstin Lanke, Wouter Graumans, Geert-Jan van Gemert, Karina Teelen, Rianne Siebelink-Stoter, Marga van de Vegte-Bolmer, Cornelus C Hermsen, Methodology, Writing—review and editing; Lisette Meerstein-Kessel, Data curation, Methodology, Writing—review and editing; Quirijn de Mast, André J van der Ven, Investigation, Writing—review and editing; Karen Ivinson, Resources, Validation, Project administration, Writing—review and editing; Katharine A Collins, Conceptualization, Writing—review and editing; Christian F Ockenhouse, James McCarthy, Conceptualization, Resources, Writing—review and editing; Robert W Sauerwein, Conceptualization, Supervision, Investigation, Writing—review and editing; Teun Bousema, Conceptualization, Data curation, Formal analysis, Supervision, Funding acquisition, Investigation, Methodology, Writing—original draft

## Author ORCIDs

Isaie J Reuling http://orcid.org/0000-0002-1783-0735
Wouter Graumans http://orcid.org/0000-0003-3952-6491
John Bradley http://orcid.org/0000-0002-9449-4608
Teun Bousema http://orcid.org/0000-0003-2666-094X

## Ethics

Clinical trial registration NCT02836002
Human subjects: Informed consent was provided by all participants at screening visit. The central committee for research involving human subjects (CCMO), and the Western Institutional Review Board (WIRB) approved the protocol for this study (NL56659.091.16). The trial was conducted according to the principles outlined in the Declaration of Helsinki and Good Clinical Practice standards, and registered at ClinicalTrials.gov, identifier NCT02836002.

## Decision letter and Author response

Decision letter https://doi.org/10.7554/eLife.31549.sa1
Author response https://doi.org/10.7554/eLife.31549.sa2

# Additional files

## Supplementary files

• Source code 1. R codes used for Bayesian statistical analysis.

• Source code 2. SAS code used for estimation of gametocyte half-life for gametocytes.

• Supplementary file 1. Individual data of the participants included in analysis.

• Supplementary file 2. Selected *P. falciparum* gene targets and primers of qRT PCR assays.

• Supplementary file 3. Quality parameters of qRT PCR and qPCR. The table shows for each target: limit of detection (LOD; defined as lowest pathogen concentration with reproducible detection); limit of quantification (LOQ; defined as lowest pathogen concentration where the CV was <5%), slope, efficiency (E), and the coefficient of correlation of combined trendlines ($R^2$).

• Supplementary file 4. Clinical trial protocol.

• Reporting standard 1. CONSORT extension for Pilot and Feasibility Trials Checklist.

• Transparent reporting form

## Data availability

The following dataset was generated:

| Author(s) | Year | Dataset title | Dataset URL | Database and Identifier |
|---|---|---|---|---|
| Reuling IJ, Bousema T, Sauerwein RW | 2017 | Data from: Induction of Plasmodium falciparum gametocytemia in the Controlled Human Malaria Infection model: a randomised trial comparing four antimalarial drug regimens | http://dx.doi.org/10.5061/dryad.60h41 | Available at Dryad Digital Repository under a CC0 Public Domain Dedication, 10.5061/dryad.60h41 |

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
