## [Decision Letter]

Thank you for submitting your article "Induction of *Plasmodium falciparum* gametocytemia in the Controlled Human Malaria Infection model: a randomised trial comparing four antimalarial drug regimens" for consideration by *eLife*. Your article has been favorably evaluated by Prabhat Jha (Senior Editor) and three reviewers, one of whom, Ben Cooper (Reviewer #3), is a member of our Board of Reviewing Editors. The following individual involved in review of your submission has agreed to reveal their identity: Nicholas J White (Reviewer #2).

The reviewers have discussed the reviews with one another and the Reviewing Editor has drafted this decision to help you prepare a revised submission.

Summary:

This paper describes a novel controlled human malaria infection model for *Plasmodium falciparum* that consistently and safely induces gametocytaemia using four different drug-regimens based on sulfadoxine-pyrimethamime and piperaquine. Estimated mean gametocyte circulation times were about 3 and 5 days for male and female gametocytes respectively and the timing of gametocyte appearance indicated probable commitment to gametocyte production in the first wave of asexual parasites. This work has importance for understanding gametocyte dynamics and evaluating transmission blocking interventions.

Essential revisions:

1) Clarification of sample size calculations is required (see reviewer 1 comments).

2) The section on parasite DNA and RNA quantification needs strengthening.a) The performance characteristics of the 18S DNA and mRNA assays needs presenting in detail or references provided to detailed validation – accuracy at different densities, reproducibility, linearity, criteria for limits of detection etc.

b) The volume of blood taken and assayed needs stating.

c) Validation of gametocyte quantification needs support – particularly with reference to stability of transcript numbers per cell and any assumptions made in the derivation of a gametocyte density (see reviewer 2 comments).

3) Details of statistical analysis performed and reporting of results of the analysis are inadequate. See comments from all three reviewers for more details. The response should include addressing reviewer 2's concerns about modelling the decline in gametocyte densities. It would also be helpful for authors to provide the code used for data analysis as recommended in *eLife*'s transparent reporting form.

4) The CONSORT guidelines for trial reporting should be followed (see reviewer 1 comments).

5) More details of the membrane feeding experiments need to be reported (see reviewer 1 and 3 comments).

6) Figures 2, 4 and (ideally) 5 should be improved to better convey information (see reviewer 1 and 3 comments).

*Reviewer #1:*

Deliberate infections treated with SP and piperaquine have been shown to be followed by gametocytaemia. This paper reports a small study with 16 participants with the objective of determining which of four drug combinations work best for inducing gametocytaemia in CHMI.

Introduction, last paragraph. The aim is to "induce stable gametocyte carriage". What is considered to be stable is buried as a footnote to Supplementary file 1. The optimal characteristics (density, duration) are also implied rather than stated.

The primary study outcome (subsection “Study outcome”) is prevalence of gametocytes by Pfs25 qRT-PCR. Was this the presence of female gametocytes at any of the twice daily measurements from day 6?

The sample size calculation (subsection “Statistical analysis”, first paragraph) appears to set the bar very low. Even so, I could not quite reproduce the calculation: an exact binomial 90% confidence interval for 6/7 would have lower bound 47.6%. It might be that you used an approximation, but since the numbers are small an exact CI seems more appropriate. Why did you choose a 90% confidence interval rather than the more conventional 95%? (with 95% CI, the lower bound for 6/7 would be 42.1%). The numbers are very small, and nowhere is it really explained why more data would be hard to collect.

There was an odd dissonance between the simple comparisons for hypothesis tests (ANOVA, Fisher's Exact test) and a more sophisticated analysis to "identify which study arm potentially deviated from the others". I'm not sure what this involves since few details were given, but it includes a prior for R^2 which seems unintuitive, and, given the limited sample size, could be stretching the data further than it can bear. It was also written without clarity in the Results e.g. "96.5% probability of being the highest" – the LD-PIP/SP mean was anyway the highest in the trial. Do you mean the chance of the population mean of LD-PIP/SP being the highest?

It would be useful to briefly state the assumptions in the previously developed model to estimate the gametocyte half-life.

It is not obvious that the AUC is the most useful measure – a short spike in gametocytes could be equal to longer low-level gametocytaemia. Clarification of what the optimal characteristics for drug development are would help. The AUC as a measure may also suffer from the non-independence of each time-point from the previous time-points – from a high peak (which may have occurred by chance or not due to the drugs), then further high values might tend to follow. A suggestion only: it might be possible to take into account the non-independence between time-points and use all the data, by using a statistical time-series model with a lag from the asexual densities or a simple mechanistic model fitted to the data. You might need more than four participants per arm for that but you might also be able to borrow strength for some parameters.

The Results section focuses on differences between the arms but it is unfortunate that the AUC is the only reported measure of gametocytaemia which is (borderline) significant between the arms for gametocytaemia (or the other p-values are just not mentioned). Looking at the graphs, I think the effect is plausible, but there is very limited evidence due to the small sample size.

That three mosquitoes were infected cannot be interpreted without the numbers of mosquitoes fed (Results, fifth paragraph).

The male and female gametocytes were measured by different assays which may have different levels of detection. The results for the sex-ratio should be interpreted with caution.

The CONSORT guidelines for reporting should be followed for *eLife*. Since this is a phase 1/2 trial with a small sample of healthy participants, the most appropriate CONSORT guidelines would be: CONSORT 2010 statement: extension to randomised pilot and feasibility trials (http://www.bmj.com/content/355/bmj.i5239). The reporting guidelines cover design, randomization, intervention, sample size, bias, generalisability and access to the pilot protocol.

Given that the emphasis in the results is on the differences between arms, Figure 4, where all groups are combined, is not easy to interpret.

There is little mention of the next steps for the induced gametocytes. If they are to be used in testing interventions that reduce gametocyte development, how would they be used?

It is not discussed why the different combinations of drug might have different effects.

*Reviewer #2:*

This is a very interesting and informative study of *Plasmodium falciparum* malaria gametocyte dynamics which exploits the opportunities provided by the resurgence of interest in human challenge studies and the development of methodologies for sensitive quantitation of nucleic acid concentrations. In general this is a very good piece of work but more details on the validation of qPCR gametocyte quantitation are essential if it is to be published. If there is one disappointment it is the complete absence of reference to the early observations of gametocytaemia in human challenge studies – notably the work of Shute and Cuica whose conclusions were in broad agreement with the current paper. The earliest reference is from 1986.

A list of questions or comments in the order they appear in the manuscript:

1) "It is widely accepted that malaria elimination is unlikely to be attainable in the majority of endemic settings with currently available resources and tools". Perhaps this could be attenuated? Many think the obstacles are primarily political, organisational and financial.

2) "maturation of gametocytes takes place predominantly in the bone marrow".

3) The 3D7 "lineage" is well known, but it is also well known to be different in different laboratories! Perhaps a few additional sentences on this particular lineage would be valuable. In particular whether there is any evidence that serial passage through volunteers and these laboratory reared mosquitoes has altered its biological properties – notably infectivity over the years? Please also confirm wild type PfDHFR and DHPS and single copy plasmepsin 2/3.

4) "until malaria parasites were detected at a density of {greater than or equal to}5000 parasites per milliliter (Pf/mL) by qPCR or a positive thick blood smear" – – were any volunteers symptomatic at this density?

5) Piperaquine base or piperaquine phosphate?

6) Safety seems rightly to have been a major concern. Did anyone look at the ECG if metoclopramide was given to piperaquine recipients? Was there any additional QTc prolongation?

7) The section on parasite DNA and RNA quantitation needs considerable strengthening as it is the central component of the paper. This is important.a) The performance characteristics of the 18S DNA and mRNA assays needs presenting in detail or references provided to detailed validation – accuracy at different densities, reproducibility, linearity, criteria for limits of detection etc.

b) The volume of blood taken and assayed needs stating.

c) Validation of gametocyte quantitation needs support – particularly with reference to stability of transcript numbers per cell and any assumptions made in the derivation of a gametocyte density.

8) Bousema et al., 2010 is referred to – which if I understand correctly assumes a single first order decline in gametocyte densities. Is this justified? It would be valuable to describe, if possible, the residuals around the model fits and comment on any heteroscedasticity. This is particularly important considering that a major finding of this report is that male gametocytes appear to be cleared more rapidly than female gametocytes. If in fact the gametocyte clearance profile is more complex (e.g. multiexponential) then the lower density male gametocytes may appear to be more rapidly eliminated whereas, in fact, the slower terminal phase of elimination is below the limit of accurate detection. A similar problem has bedevilled assessment of the pharmacokinetic properties of slowly eliminated antimalarials.

9) It is not very clear what modelling approach was used to fit the model – was this a mixed effect model? Were any covariates incorporated – if so which? What programme was used?

10) Is there any explanation for the liver function test abnormalities?

11) "SP has long been associated with a rapid appearance of gametocytes that is too early to be explained by de novo gametocyte production upon drug pressure and has thus been hypothesized to reflect an efflux of sequestered gametocytes upon treatment". True – but it is not very clear from the text whether this study's results supports this hypothesis. Could the authors be explicit here? The implication is that such early released gametocytes should be more immature- and thus the period during which *Plasmodium falciparum* gametocytes are not infectious (after release into the circulation) should be longer in these circumstances. Is there any evidence for this?

12) "The highly abundant Pfs25 mRNA makes the female gametocyte qRT-PCR more sensitive than the male PfMGET qRT-PCR." Could this be elaborated upon- or at least referenced?

*Reviewer #3:*

This paper describes a novel controlled human malaria infection model that consistently achieves gametocyte carriage. This work is important with implications for understanding of *P. falciparum* transmission dynamics and broad significance for the future study of transmission blocking interventions.

The paper is, in most respects, clearly written. There are, however, some aspects of reporting where it is not entirely clear exactly what was done and why (particularly in relation to statistical methods).

Results, fifth paragraph. Details of these membrane feeding experiments are lacking from the Materials and methods. In particular, how many mosquitos on each feeding day? Even if full details are presented in the references describing the protocols for these experiments, it would be useful to summarise some of the key numbers in the manuscript to give the reader an idea of what these numbers mean. See also Discussion, third paragraph – "the very low rate". Since only the numerators are reported, there is no information on what this rate actually is.

Subsection “Statistical analysis”, second paragraph. More details need to be given here. What non-parametric tests were used and what exactly was the statistical model implemented in the Bayesian framework (code for this does not appear to have been provided). Also, the R package *rstanarm* is mentioned, but appears not to be acknowledged in the references. It probably should be along with the appropriate references for Stan and R.

Table 3 is confusing. Why are some adverse events listed twice in the leftmost column? Duration time units should be given.

---

## [Author Response]

Essential revisions:1) Clarification of sample size calculations is required (see reviewer 1 comments).

We have clarified the sample size calculation in response to the comments of reviewer 1. We used Wilson confidence intervals and have explained this in more detail in the comment of reviewer 1 and the revised manuscript.

2) The section on parasite DNA and RNA quantification needs strengthening.a) The performance characteristics of the 18S DNA and mRNA assays needs presenting in detail or references provided to detailed validation – accuracy at different densities, reproducibility, linearity, criteria for limits of detection etc.

We have provided all the requested details of our assay performance in the supporting information of the revised manuscript. We agree that this is a considerable improvement of the original data presentation and important for the reader. Please find our detailed response to this question below in the section related to comments of reviewer 2.

b) The volume of blood taken and assayed needs stating.

Similarly, we have provided details on blood volume taken, used for extraction and pathogen quantification. The detailed response is given below in the section of reviewer 2.

c) Validation of gametocyte quantification needs support – particularly with reference to stability of transcript numbers per cell and any assumptions made in the derivation of a gametocyte density (see reviewer 2 comments).

We have provided these details using new data related to the current manuscript and a re-analysis of data of a recently published study where we originally presented the methodology and determined the stability of gametocyte transcripts in maturing male and female gametocytes (Stone et al., 2017). We were also able to provide details on the number of Pfs25 transcripts per female gametocyte and the number of PfMGET transcripts per male gametocyte.

3) Details of statistical analysis performed and reporting of results of the analysis are inadequate. See comments from all three reviewers for more details. The response should include addressing reviewer 2's concerns about modelling the decline in gametocyte densities. It would also be helpful for authors to provide the code used for data analysis as recommended in eLife's transparent reporting form.

We have provided the details on the statistical analysis methods used and the reporting of the analysis (see reviewer comments below). In addition, the annotated codes used for data analysis are now added as supplementary information in the manuscript.

4) The CONSORT guidelines for trial reporting should be followed (see reviewer 1 comments).

We have added the CONSORT extension for Pilot and Feasibility Trials Checklist to the dossier, and adjusted the manuscript as per CONSORT guidelines.

5) More details of the membrane feeding experiments need to be reported (see reviewer 1 and 3 comments).

Membrane feeding experiments were an exploratory outcome in the current study and in the initial submission we provided limited details in order to focus on the primary and secondary endpoints. In the revision, we provide all details on the Direct Membrane Feedings Assays (DMFA) that was performed on days 21, 25 and 31 post-infection with ~300 mosquitoes per feed per participant (total of ~14.400 mosquitoes). This information has been added to the Materials and methods and Results sections.

6) Figures 2, 4 and (ideally) 5 should be improved to better convey information (see reviewer 1 and 3 comments).

We have adjusted the figures in the manuscript to accommodate the requests of reviewers 1 and 3. Please also find our responses below.

Reviewer #1:[…] Introduction, last paragraph. The aim is to "induce stable gametocyte carriage". What is considered to be stable is buried as a footnote to Supplementary file 1. The optimal characteristics (density, duration) are also implied rather than stated.

We agree that this was not clear in the original manuscript. We did not define ‘stable gametocytaemia’ a priori. We have thus modified our wordings in the Materials and methods section on the calculation of gametocyte positive days, peak gametocyte density and area under the curve of gametocyte density versus time. In the Discussion section, we comment on the optimal characteristics of the model, including stable gametocyte carriage, for different use scenarios and the extent to which the current data achieve these optimal characteristics.

The primary study outcome (subsection “Study outcome”) is prevalence of gametocytes by Pfs25 qRT-PCR. Was this the presence of female gametocytes at any of the twice daily measurements from day 6?

The reviewer is correct in this assumption. The prevalence of gametocytes was defined as the presence of female gametocytes as measured by qRT-PCR targeting female-specific Pfs25 mRNA at any of the twice daily measurements from day 6. This information has been added to the Materials and methods subsection “Study outcome”.

The sample size calculation (subsection “Statistical analysis”, first paragraph) appears to set the bar very low. Even so, I could not quite reproduce the calculation: an exact binomial 90% confidence interval for 6/7 would have lower bound 47.6%. It might be that you used an approximation, but since the numbers are small an exact CI seems more appropriate. Why did you choose a 90% confidence interval rather than the more conventional 95%? (with 95% CI, the lower bound for 6/7 would be 42.1%). The numbers are very small, and nowhere is it really explained why more data would be hard to collect.

The sample size calculation was used to plan the study and ensure we had sufficient power to determine whether the majority of volunteers would develop detectable levels of mature gametocytes upon treatment. Controlled Human Malaria Infection (CHMI) studies are logistically challenging and this CHMI study was particularly challenging in terms of the number of study participants that were examined in exploratory mosquito feeding assays. As a consequence, the number of volunteers was informed by both feasibility and study power and a 90% confidence interval was chosen to find a balance between feasibility and the precision in study endpoints.

We used a Wilson confidence interval in the planning phase of this study. The Wilson confidence interval is particularly informative in case of a small number of observations. The Wilson confidence interval is considered an improvement over the normal approximation interval in that the actual coverage probability is closer to the nominal value. We used the command ‘cii 7 6, wilson level(90)’ in STATA, giving the confidence limit of 54.8 if we assumed 6/7 volunteers per arm would develop gametocytes. A 95% CI would have given a limit of 48.7. We have clarified this in the revised Materials and methods section.

Whilst our sample size calculations were optimistic, our study data indicate that this optimism was justified with 100% of all volunteers developing mature gametocytes (16/16, 90% Wilson CI 85.5-100% or 95% CI 80.6-100%.

We have further clarified the limitations of our modest sample size in the Discussion section, especially notable when comparing study arms.

“In the current study, group sizes are limited and comparisons between treatment arms have to be interpreted with caution. […] Our sample size calculation was based on the optimistic assumption that the vast majority of volunteers would develop mature gametocytes; an assumption that was supported by the current data. With our limited study size, our findings indicate…”

There was an odd dissonance between the simple comparisons for hypothesis tests (ANOVA, Fisher's Exact test) and a more sophisticated analysis to "identify which study arm potentially deviated from the others". I'm not sure what this involves since few details were given, but it includes a prior for R^2 which seems unintuitive, and, given the limited sample size, could be stretching the data further than it can bear. It was also written without clarity in the Results e.g. "96.5% probability of being the highest" – the LD-PIP/SP mean was anyway the highest in the trial. Do you mean the chance of the population mean of LD-PIP/SP being the highest?

We have clarified our statistical methods in the revised manuscript. There was no statistically significant difference in time to appearance of gametocytes between the study arms. (p = 0.26). A simple ANOVA for gametocyte density (AUC) did however suggest a statistical significant difference between arms (p = 0.04). Following this conventional frequentist analysis, we used a Bayesian model to determine whether there was evidence for a difference between study arms. With this Bayesian model, this difference was identified as being caused by the LD-PIP/SP arm, which had significantly higher mean gametocyte density (in terms of AUC) than the other three arms. Because the model is Bayesian, we are actually able to calculate the probability that this conclusion is true (i.e. the posterior probability) given the model, which we calculated to be 94.4% (94.0% after correction for asexual density AUC). Regarding the “unintuitive” prior for R^2, we did not add any prior information to the Bayesian model, which in the case of standard linear regression in the package *rstanarm* takes its prior information in the form of a prior for R^2. Essentially, this means that we allowed model parameters to take on any value and that these values were solely informed by the data.

We now also provide further details in the Results section.

“The LD-PIP/SP arm had significantly higher gametocyte concentrations (area under the curve, AUC) than each of the other three arms with a posterior probability of 99.1% (compared to the LD-SP/SP arm), 98.9% (LD-SP/PIP), and 95.4% (LD-PIP/PIP), respectively. […] After correction for the asexual AUC, the probabilities of the gametocyte AUC in the LD-PIP/SP arm being higher than the other three decreased to 97.2%, 96.3%, and 96.2%, and the probability of LD-PIP/SP being higher than all other groups decreased to 94.0%.”

In addition, we expanded our Bayesian analysis to also look at the number of gametocyte positive days per individual by means of a multi-level logistic regression model (including a random effect for individual deviations from the four group means). The model was fitted to data on the number of days that each individual was tested using for Pfs25mRNA (denominator) and the number of days that individuals tested positive for gametocytes by Pfs25 qRT-PCR (≥ 5 gametocytes/mL) at least once (numerator). Consistent with the results from the AUC analysis, the average proportion of days that individuals in each group tested positive for Pfs25 mRNA was estimated at 27.4% (LD-SP/SP), 35.9% (LD-SP/PIP), 51.4% (LD-PIP/PIP), and 48.3% (LD-PIP/SP). The LD-PIP/PIP and LD-PIP/SP arms (i.e. those receiving “low dose PIP”) each had significantly higher average proportions of gametocyte-positive days than both arms LD-SP/SP and LD-SP/PIP (posterior probability 90.8% and 86.1%, respectively; 81.1% joint probability of arms LD-PIP/PIP and LD-PIP/SP both being higher than both LD-SP/SP and LD-SP/PIP). This information has been added to the Results.

It would be useful to briefly state the assumptions in the previously developed model to estimate the gametocyte half-life.

We have provided the requested details in the revised supplemental information and expanded the Materials and methods section. The model was a mixed effects model that had no covariates (other than time) fit in SAS 9.4. The annotated model code, including all assumptions are now provided in detail.

It is not obvious that the AUC is the most useful measure – a short spike in gametocytes could be equal to longer low-level gametocytaemia. Clarification of what the optimal characteristics for drug development are would help. The AUC as a measure may also suffer from the non-independence of each time-point from the previous time-points – from a high peak (which may have occurred by chance or not due to the drugs), then further high values might tend to follow. A suggestion only: it might be possible to take into account the non-independence between time-points and use all the data, by using a statistical time-series model with a lag from the asexual densities or a simple mechanistic model fitted to the data. You might need more than four participants per arm for that but you might also be able to borrow strength for some parameters.

We appreciate these comments that make a case for a more comprehensive model of gametocyte commitment, maturation and longevity as a function of asexual parasitaemia. This is a longer term objective of our group once we have optimized our model that will require a larger number of participants to accurately parameterize gametocyte commitment and release rates. For the current purposes (i.e. to establish the CHMI transmission model, to select an optimal treatment regimen and to determine the time of first appearance of gametocytes), we consider the presented analyses appropriate.

For the comparison of drug regimens, we consider the AUC a useful summary measure as gametocyte density patterns looked very similar between individuals (i.e. similar levels of variation over time and not different types of patterns of variation, e.g. “short spikes” vs. “continuous stable levels”).We have, however, also incorporated analyses on the number of gametocyte positive days (showing the same patterns between groups as the AUC) and provide a more detailed presentation of individual parasite and gametocyte curves.

Importantly, and this was not clearly described in the original submission, a future study will examine infectivity to mosquitoes in more detail. For that, the current data are crucial since they inform us on the time till gametocyte appearance and the plateau phase in gametocyte density. A short spike in gametocytes (without a plateau phase of gametocyte densities that are sufficiently high to measure gametocyte clearance or infectivity) would be operationally very challenging since it would be nearly impossible to define the optimal moment for mosquito feeding assays and/or initiation of gametocytocidal drugs. This is clarified in the revised Discussion section.

The reviewer’s concern of correlation over time within individuals would indeed be an issue when one would consider the observed densities at individual time points as independent identically distributed variables. For this specific reason, we reduced the time series data to one data point per individual (i.e. the AUC), which essentially relaxes the assumption of inter individual differences observations. So in our analysis, despite the fact that the AUC may be higher in some individuals by shear chance (which the model captures in the error term of the linear predictor), the posterior probability that the AUC is highest in the LD-PIP/SP arm on average is still a very convincing 94.4% (if groups where identical, one would expect a posterior probability of about 1 / 4 = 25%).

The Results section focuses on differences between the arms but it is unfortunate that the AUC is the only reported measure of gametocytaemia which is (borderline) significant between the arms for gametocytaemia (or the other p-values are just not mentioned). Looking at the graphs, I think the effect is plausible, but there is very limited evidence due to the small sample size.

As indicated above, we have clarified the limitations of our study size in the Discussion section. We have further extended our analysis to the proportion of days that individuals test positive for gametocytes, showing that individuals in study arms LD-PIP/PIP and LD-PIP/SP on average experience more gametocyte-positive days, with little difference between LD-PIP/PIP and LD-PIP/SP. The previous AUC analysis then further shows that individuals in LD-PIP/SP have higher gametocyte concentrations than in all other groups.

We would also like to clarify that the finding of 94.4%% probability is not borderline as it represents a posterior probability, i.e. the probability that there really is a difference between groups. In a Bayesian context, if in reality there is no real difference between four groups (and this is reflected by the data), the posterior probability of one group being the highest would be estimated around 25%. Therefore, our estimate of 94.4% probability of group 4 being the highest is very convincing.

That three mosquitoes were infected cannot be interpreted without the numbers of mosquitoes fed (Results, fifth paragraph).

We have now provided full details of this exploratory endpoint. Expressed as a proportion of all examined mosquitoes, 0.0002% (3/14400) of mosquitoes became infected in these exploratory assessments.

The male and female gametocytes were measured by different assays which may have different levels of detection. The results for the sex-ratio should be interpreted with caution.

We fully agree. We have now provided full details on our qRT-PCR performance, sensitivity and differences in sensitivity between arms. We have also rephrased our results on the sex ratio to highlight uncertainties with these estimates that rely on a ratio of two separate qRT-PCRs.

The CONSORT guidelines for reporting should be followed for eLife. Since this is a phase 1/2 trial with a small sample of healthy participants, the most appropriate CONSORT guidelines would be: CONSORT 2010 statement: extension to randomised pilot and feasibility trials (http://www.bmj.com/content/355/bmj.i5239). The reporting guidelines cover design, randomization, intervention, sample size, bias, generalisability and access to the pilot protocol.

We thank the reviewer for its comment and have added the CONSORT extension for Pilot and Feasibility Trials Checklist to the dossier, and adjusted the manuscript as per CONSORT guidelines. In addition, the clinical trial protocol has been added to the submission.

Given that the emphasis in the results is on the differences between arms, Figure 4, where all groups are combined, is not easy to interpret.

We appreciate this comment. Figure 4 represents the total female and male gametocyte density of all participants to give an impression of the female-biased sex-ratio per time-point and its time course. Figure 4—figure supplement 1, shows the individual PCR data per study arm. In addition, we have now provided an illustration of male and female gametocyte clearance dynamics for individual study participants (Figure 4—figure supplement 2).

There is little mention of the next steps for the induced gametocytes. If they are to be used in testing interventions that reduce gametocyte development, how would they be used?

We have now better explained this in the Discussion section. We describe the ideal gametocyte characteristics for different use scenarios of the model (e.g. gametocytocidal drugs or transmission-blocking vaccines) and end the Discussion section with a paragraph on future scenarios.

“For the evaluation of gametocytocidal interventions in the CHMI transmission model, gametocyte densities should be sufficiently high to quantify an intervention-associated reduction in gametocyte appearance or gametocyte half-life. […] Low infectivity in membrane feeding assays may be overcome by achievement of higher gametocyte densities in the model, and the use of gametocyte concentration methods (Reuling et al., 2017), or by direct skin feeding assays (Bousema et al., 2012).”

And:

“Here, we present a novel CHMI transmission model for *P. falciparum* that can be used to study gametocyte biology and dynamics providing novel insights and tools in malaria transmission and elimination efforts. […] The current work lays the foundation for fulfilling the critical unmet need to evaluate transmission-blocking interventions against falciparum malaria for downstream selection and clinical development.’

It is not discussed why the different combinations of drug might have different effects.

We have clarified this in the revised Discussion section.

“With our limited study size, our findings indicate that none of the study drugs prevented the appearance of gametocytes after treatment, thereby suggesting limited or no effect of PIP and SP on developing or mature gametocytes (Bolscher et al., 2015). […] One hypothesis that may explain differences between study arms would be similar gametocyte commitment in all arms after T1 but a more rapid release of gametocytes that accumulated in the bone marrow between T1 and T2.”

Reviewer #2:This is a very interesting and informative study of Plasmodium falciparum malaria gametocyte dynamics which exploits the opportunities provided by the resurgence of interest in human challenge studies and the development of methodologies for sensitive quantitation of nucleic acid concentrations. In general this is a very good piece of work but more details on the validation of qPCR gametocyte quantitation are essential if it is to be published. If there is one disappointment it is the complete absence of reference to the early observations of gametocytaemia in human challenge studies – notably the work of Shute and Cuica whose conclusions were in broad agreement with the current paper. The earliest reference is from 1986.

We appreciate this comment. One of the joys of malaria research is that there is a wealth of excellent early data and we agree that this deserves referencing and attention in the manuscript. We have now added these references in the manuscript and described some of this excellent early work in detail in the Introduction., e.g.:

“Early work based on the microscopic evaluation therapeutic controlled *P. falciparum* infections reported that gametocytes may make their appearance in small numbers around 10 days following the first day of fever (Shute and Maryon, 1951; Ciuca, Chearescu and Lavrinenko, 1937).”

A list of questions or comments in the order they appear in the manuscript:1) "It is widely accepted that malaria elimination is unlikely to be attainable in the majority of endemic settings with currently available resources and tools". Perhaps this could be attenuated? Many think the obstacles are primarily political, organisational and financial.

We agree with the reviewer and have adjusted the sentence:

“Novel interventions may support malaria elimination efforts in endemic settings (Griffin et al., 2010) that are further dependent on political and financial commitments to maximize coverage with currently available interventions and improve surveillance systems to optimize disease notification and treatment (Moonen et al., 2010).”

2) "maturation of gametocytes takes place predominantly in the bone marrow".

We agree with the reviewer and have adjusted this sentence:

“maturation of gametocytes takes place predominantly in the bone marrow”.

3) The 3D7 "lineage" is well known, but it is also well known to be different in different laboratories! Perhaps a few additional sentences on this particular lineage would be valuable. In particular whether there is any evidence that serial passage through volunteers and these laboratory reared mosquitoes has altered its biological properties – notably infectivity over the years? Please also confirm wild type PfDHFR and DHPS and single copy plasmepsin 2/3.

The 3D7 lineage that was used in the current study is based on a 3D7 bank described in detail in Cheng et al., AJTMH 1997. The material has thus not been in constant passage. Whilst differences in infectivity exist between parasite lines (McCall, Sci Transl Med 2017 as recent example), the material we used was generating gametocyte densities not dissimilar from our routine NF54 line and we achieved high mosquito infection rates and oocyst and sporozoite densities for inoculation in the current study.

We have clarified the origin of the 3D7 material, including the original reference, in the revised manuscript. In addition, we used available Illumina Whole genome sequencing data for this lineage (https://www.ebi.ac.uk/ena/data/view/PRJEB12838). Reads were aligned to the *P. falciparum* reference genome v3 (plasmoDB) with bowtie2 (sourceforge) and consensus sequences for dhps and dhfr genes were obtained with samtools. Read depth (coverage) was assessed using bedtools. With this approach, we determined dhfr/dhps mutations in our 3D7 parasites; only one mutated locus in the dhps gene was found (PfDHPS A437G), which is not associated with treatment failure as single mutation (Staedke et al. 2004). WGS analysis thus rules out sulfadoxine-pyrimethamine drug resistance of the 3D7 lineage we used. Plasmepsin II/III duplication events are associated with piperaquine resistance (Witkowski et al., 2017) but were not found in the 3D7 WGS data. However, as the neighboring genes Plasmepsin I and IV have sequence similarities, unambiguous quantification is challenging and a final assessment would involve more specific targeting, e.g. by qPCR.

**gene****locus****3D7 codon****amino acid****mutated****DHPS**A437GGGTGyes**DHPS**K540EAAAKno**DHFR**N51IAATNno**DHFR**C59RTGTCno**DHFR**S108NAGCSno

Importantly, piperaquine sensitivity of the 3D7 lineage was previously confirmed by in vivo experiments (Pasay et al., 2016).

We included the following information in the revised manuscript in the section describing the parasite lineage:

“The 3D7 lineage that was used in the current study is based on a 3D7 bank described in detail in Cheng et al., (Cheng et al., 1997). […] We conclude that the lineage used was sensitive to both sulfadoxine-pyrimethamine and piperaquine.”

4) "until malaria parasites were detected at a density of {greater than or equal to}5000 parasites per milliliter (Pf/mL) by qPCR or a positive thick blood smear" – were any volunteers symptomatic at this density?

The majority of the volunteers became symptomatic (1-2 day) before the initiation of treatment. As expected from previous CHMI studies the majority of symptoms were experienced after the first treatment. The time course of adverse events is shown in Figure 6B, and Supplementary file 1. shows the day of the first treatment per volunteer.

5) Piperaquine base or piperaquine phosphate?

We have now added the full nonproprietary name of piperaquine phosphate to the Materials and methods – subsection “Procedures”.

6) Safety seems rightly to have been a major concern. Did anyone look at the ECG if metoclopramide was given to piperaquine recipients? Was there any additional QTc prolongation?

Piperaquine recipients received an ECG, 4-12 hours after treatment of each dose of piperaquine (based on the expected maximal piperaquine concentrations after oral dosing). The maximum dose of piperaquine used in our study is approximately only 1/3 of the curative dose used in the dihydroartemisinin piperaquine combination therapy (label use), therefore, the risk of QT-prolongations in this study, next to administration on an empty stomach (to reduce the peak concentrations) is assessed to be very low. The combination of the very rarely described QT-prolongations with metoclopramide and the sporadic, and mostly single-dose (10mg) metoclopramide use in our study participants made an additional risk of QT-prolongations highly unlikely. Therefore, additional ECG assessments after metoclopramide were not deemed necessary at discretion of the investigators. No QT-prolongations were found after or between piperaquine treatments in this study.

7) The section on parasite DNA and RNA quantitation needs considerable strengthening as it is the central component of the paper. This is important.a) The performance characteristics of the 18S DNA and mRNA assays needs presenting in detail or references provided to detailed validation – accuracy at different densities, reproducibility, linearity, criteria for limits of detection etc.

We appreciate this comment and have provided all details of our molecular assay performance in the revised manuscript and supporting information. We agree that this is essential, especially since the lower sensitivity of the male-specific PfMGET qRT-PCR is relevant for the interpretation of the study results. Our assay characteristics that are now also included in the revision (Figure 5, Supplementary file 2, Supplementary file 3, Figure 5—figure supplement 1, Figure 5—figure supplement 2). The limit of detection was defined as the lowest concentration where all trendline samples were still positive, the limit of quantification was defined as the lowest concentration where the coefficient of variation was <15% and where duplicate study samples showed good agreement (Figure 5—figure supplement 2).

b) The volume of blood taken and assayed needs stating.

We have now added more information on the molecular assays used in the study, including the volume of blood used per assay.

“For the quantification of the *P. falciparum* Pfs25 transcript levels total NA was RQ1 DNaseI treated according to the manufacturer’s protocol. […] Detailed information on the validation and performance characteristics of the assays can be found in the supporting materials (Figure 5; Supplementary file 2, 3; Figure 5—figure supplement 1, 2).”

*c) Validation of gametocyte quantitation needs support – particularly with reference to stability of transcript numbers per cell and any assumptions made in the derivation of a gametocyte density.*

Full details on the stability of transcripts are now provided in the supplemental information to the revised manuscript. In our original presentation of the methodology (Stone et al., 2017) we determined the stability of transcripts in maturing male and female gametocytes. Culturing of NF54 was done without prior gametocyte sex-sorting and female (top panel) and male (bottom panel) transcripts were quantified in the same source culture material using Illumina RNAseq with paired reads, 37bp in length. Transcript counts after normalization for library size are presented for d4 till d16 after synchronization; stage IV gametocytes were observed on day 5 and stage V gametocytes on day 7. These results indicate that for both male and female markers transcript abundance is low in stage IIb gametocytes and peaks in stage V gametocytes after which transcript numbers appear stable (Pfs25) or may decline at late time-points (PfMGET).

8) Bousema et al., 2010 is referred to – which if I understand correctly assumes a single first order decline in gametocyte densities. Is this justified? It would be valuable to describe, if possible, the residuals around the model fits and comment on any heteroscedasticity. This is particularly important considering that a major finding of this report is that male gametocytes appear to be cleared more rapidly than female gametocytes. If in fact the gametocyte clearance profile is more complex (e.g. multiexponential) then the lower density male gametocytes may appear to be more rapidly eliminated whereas, in fact, the slower terminal phase of elimination is below the limit of accurate detection. A similar problem has bedevilled assessment of the pharmacokinetic properties of slowly eliminated antimalarials.

The reviewer is correct that the model assumes a single first order decline in male gametocyte densities. The decline is on the linear scale; but the error structure is a Tobit distribution on the log scale (the SAS code is now provided in the supplemental information). The fact that the error structure is on the log scale means that the heteroscedasticity is less likely to be a problem than if the errors were on the linear scale. The residuals for the non-zero female densities is displayed in Author response image 1.

**Author response image 1. respfig1:** Model fit for CHMI-trans data female gametocytes. Log female gametocyte densities at different days of follow-up (left) and residuals of non-zero female log gametocyte densities as a function of time of follow-up (right).

The left panel shows a linear decline on the log scale consistent with the model. The right panel shows no evidence of heteroscedasticity. The residuals of non-zero measurements are larger to the right of the graph, but this reflects the fact that zero measurements are more likely in that region and only residuals for non-zero measurements are shown.

For female gametocytes 74% (98/133) of observations were non-zero so it is straightforward to assess heteroscedasticity. For males the proportion of non-zero measurements was much lower (28%, 38/135) so it is much harder to assess the fit of the model and in particular heteroscedasticity. The equivalent graph is shown in Author response image 2.

**Author response image 2. respfig2:** Model fit for CHMI-trans data male gametocytes. Log male gametocyte densities at different days of follow-up (left) and residuals of non-zero male log gametocyte densities as a function of time of follow-up (right).

The left panel of figure shows a linear decline on the log scale consistent with the model, but the smaller number of non-zero points makes it harder to assess. There is no evidence of heteroscedasticity in the right panel but there is a clear trend of larger residuals to the right of the graph. But this is explained by the fact there are more zero measurements in this region. To assess whether the model would show a better fit to the data if male gametocytes had been higher, we re-analysed data from a trial in high density gametocyte carriers in Mali (NCT02831023; Dicko et al. Lancet Infect Dis in press) that used the same PfMGET qRT-PCR. We restricted analysed to the treatment arms in this study that received non-gametocytocidal drugs dihydroartemisinin-piperaquine or amodiaquine plus sulfadoxine-pyrimethamine.

**Author response image 3. respfig3:** Model fit for male gametocytes in trial in Malian gametocyte carriers treated with non-gametocytocidal drugs dihydroartemisinin-piperaquine or amodiaquine plus sulfadoxine-pyrimethamine (NCT02831023). Log male gametocyte densities at different days of follow-up (left) and residuals of non-zero male log gametocyte densities as a function of time of follow-up (right).

In this trial, with a much larger number of non-zero values for PfMGET male gametocytes, model fit was appropriate and not different from the above model fit for Pfs25 female gametocytes.

In summary, the large number of zero measurements make it hard to assess the fit of the model to the male gametocyte densities, but we think its use if justified in the case of males by 1) The fact the model fits the female gametocyte densities well; 2) The fact the model fits the male gametocyte densities in a trial with high starting gametocyte densities well; 3) The model has been widely applied in the past to many other datasets with good fit. We accept that this model, like all models, relies on assumptions and we have made this clearer in the Discussion. We have also plotted male and female gametocyte decay curves for all volunteers with >. We observe that whilst individual gametocyte clearance lines appear to be parallel for some volunteers (Figure 4—figure supplement 2), others show evidence for a faster disappearance of male gametocytes.

9) It is not very clear what modelling approach was used to fit the model – was this a mixed effect model? Were any covariates incorporated – if so which? What programme was used?

We have now added this information in the Materials and methods section. The model was a mixed effect model that included time as only covariate. The code is now given in the supporting information so the methodology in accessible for everyone in SAS and the model assumptions are described in detail.

10) Is there any explanation for the liver function test abnormalities?

There is no clear explanation for the transient elevated transaminases in CHMI studies. There is no clear relationship between parasitemia densities and liver abnormalities, and it also seems unlikely to be directly related to anti-malarials due to variety of drugs used in CHMI. Similarly, the use of paracetamol does not support a clear relationship. Rather a combination of the above mentioned factors, and individual susceptibility may have triggered the observed abnormalities. The transient asymptomatic liver function test abnormalities are most likely directly related to the asexual parasitemia, and subsequent treatment. The exact mechanism of the transient organ injury during the malaria and its treatment is yet to be understood and ongoing work. We are currently reviewing all our CHMI data, including the many trials where treatment was initiated at lower parasite densities, and intend to write this up as a separate manuscript to describe the prevalence, severity and transient nature of liver function abnormalities upon controlled malaria infections.

11) "SP has long been associated with a rapid appearance of gametocytes that is too early to be explained by de novo gametocyte production upon drug pressure and has thus been hypothesized to reflect an efflux of sequestered gametocytes upon treatment". True – but it is not very clear from the text whether this study's results supports this hypothesis. Could the authors be explicit here? The implication is that such early released gametocytes should be more immature- and thus the period during which Plasmodium falciparum gametocytes are not infectious (after release into the circulation) should be longer in these circumstances. Is there any evidence for this?

We agree with the reviewer that early released gametocytes are possibly more immature, and therefore, possibly, less infectious. However, our preliminary mosquito infectivity results do not provide any evidence for this since its very low mosquito infection rates. We have now added additional information in the Discussion on the hypothesis of an efflux of sequestered gametocytes upon treatment:

“We hypothesized that slow acting drugs may promote the development of gametocytes (Mendez et al., 2002), potentially via microvesicles that are derived from infected erythrocytes (Nilsson et al., 2015) and differences between drug regimens in the rate at which asexual parasites are cleared upon T1 and T2 would result in different gametocyte dynamics. […] One hypothesis would be similar gametocyte commitment in all arms after T1 but a more rapid release of gametocytes that accumulated in the bone marrow between T1 and T2.’

12) "The highly abundant Pfs25 mRNA makes the female gametocyte qRT-PCR more sensitive than the male PfMGET qRT-PCR." Could this be elaborated upon- or at least referenced?

We clarified this in response to the earlier comment on assay performance.

Reviewer #3:[…] The paper is, in most respects, clearly written. There are, however, some aspects of reporting where it is not entirely clear exactly what was done and why (particularly in relation to statistical methods).Results, fifth paragraph. Details of these membrane feeding experiments are lacking from the Materials and methods. In particular, how many mosquitos on each feeding day? Even if full details are presented in the references describing the protocols for these experiments, it would be useful to summarise some of the key numbers in the manuscript to give the reader an idea of what these numbers mean. See also Discussion, third paragraph – "the very low rate". Since only the numerators are reported, there is no information on what this rate actually is.

We have now presented all data, including an estimate of the (very low) proportion of infected mosquitoes.

Subsection “Statistical analysis”, second paragraph. More details need to be given here. What non-parametric tests were used and what exactly was the statistical model implemented in the Bayesian framework (code for this does not appear to have been provided). Also, the R package rstanarm is mentioned, but appears not to be acknowledged in the references. It probably should be along with the appropriate references for Stan and R.

We have added details to the Materials and methods – statistical analysis, including references and codes used. We used a standard linear regression model (no mixed effects) for the Bayesian model for gametocyte density AUC as there was only one data point (the AUC) considered per individual. The linear regression model reported in the Results section only included trial arm as a predictor. Findings did not change when individuals’ asexual density AUC was included as predictor: the probability of the LD-PIP/SP group having a higher gametocyte density AUC was still 94.0%, compared to 94.4% when asexual density AUC was not included as a predictor.

Table 3 is confusing. Why are some adverse events listed twice in the leftmost column? Duration time units should be given.

We have adjusted Table 3 accordingly, with shading, and underlines for clarification. Furthermore, the duration time units (in days) are added to the table.